



# Role of friction terms in two-dimensional modelling of dense snow avalanches

Marcos Sanz-Ramos[1], Ernest Bladé[1], Pere Oller[2], Carlos A. Andrade[3], Glòria Furdada[4]

[1]Institut Flumen, Universitat Politècnica de Catalunya-CIMNE, Barcelona, 08034, Spain
[2]GeoNeuRisk, 08024, Barcelona, Spain
[3]Departamento de Geología, Universidad de Chile, Santiago, 8370450, Chile
[4]Departament de Geodinàmica Extrema, Universitat de Barcelona, 08028, Barcelona, Spain

*Correspondence to*: Marcos Sanz-Ramos (marcos.sanz-ramos@upc.edu)

**Abstract.** Voellmy–Salm friction model is one of the most extensively used theories for assessing the frictional terms of the
equations that describe the motion of non-Newtonian flows such as snow avalanches. Based on the Coulomb- and turbulent-
type friction, this model has been implemented in numerical tools for computation of snow avalanche dynamics based on the
Shallow Water Equations (SWE). The range of the Voellmy parameters has been discussed widely, focusing mainly on the
required values for achieving good results for the description of the moment and position of the avalanche when it stops.
However, effects of parameters on the SWE terms, and their physical interpretation have not been investigated sufficiently.
This work focuses on analysing the effects of the Voellmy–Salm parameters and cohesion on the avalanche characteristics
and evolution of the new SWE-based numerical model Iber. In the numerical scheme, an upwind discretization was used for
the solid friction and cohesion terms, while a centred one was used for the turbulent friction. Results show that the Voellmy–
Salm model dominates the avalanche dynamics and the cohesion model allows the representation of long tails, whereas the
friction and cohesion parameters may vary within a wide range.

## 1 Introduction

The growing concerns about natural hazards, in particular snow avalanches, have led to the development of ad hoc numerical
models as support tools for their analysis (Funk and Margreth, 1999; Gruber and Bartelt, 2007; Jamieson et al., 2008;
Keylock and Barbolini, 2011; Maggioni et al., 2019). Gauer et al. (2008) indicate that avalanche, from the viewpoint of their
dynamics, have been traditionally classified into dense flow avalanches and powder snow avalanches, and that often dry-
snow avalanches are not purely any of the previous types, as they consist of a powder cloud, but also a dense core below
(mixed-motion avalanches). Up to date, most numerical simulation tools available for practitioners provide results of runout
distance, flow heights, flow velocities and impact pressure for dense flow snow avalanches, or for the dense core of mixed
ones.





Snow avalanches are commonly modelled by solving mass and momentum conservation equations, which are also used for
the simulation of water flows (Shallow Water Equations, SWE, or Saint Venant equations). The SWE usually use the
Manning formula for representing the frictional terms for water flows; however, for snow, other specific models are applied.
The Voellmy–Salm friction model (Salm, 1993; Voellmy, 1955) is a popular model used to define the friction terms for
granular flows (Hussin et al., 2012; Pirulli and Sorbino, 2008; Schraml et al., 2015), also used for snow avalanches modelling
(Ancey et al., 2004; Bartelt et al., 1999; Christen et al., 2010; Keylock and Barbolini, 2011; Oller et al., 2010). This friction
model integrates the total flow resistance as the addition of two parameters, namely turbulent friction resistance and solid
phase, which are associated with flow velocity and dry-Coulomb friction, respectively. The range of these parameters has
been the object of many analysis and discussion (Ancey et al., 2004; Bartelt et al., 1999; Fischer et al., 2015; Gauer, 2014;
Gruber and Bartelt, 2007; Hussin et al., 2012; Hutter et al., 1995; Keylock and Barbolini, 2011), some of which focus on the
achievement of a good representation of the moment and position when the avalanche stops. As a result, different guidelines
and handbooks have been published (Bartelt et al., 2017; Brugnot, 2000; Christen et al., 2001; Maggioni and Gruber, 2003;
Salm et al., 1990), helping avalanche modellers to select the appropriate values of these parameters.

However, the effects of the friction model on the individual terms of the equations are commonly ignored. Thus, very
different values of the model parameters, nonphysically based, could achieve the same results on the simulation of a snow
avalanche. Additionally, using parameter values within the recommended ranges may cause difficulties in representing the
avalanche stopping. For such cases, Bartelt et al. (2017) proposed a stopping criterion based on momentum, i.e., the
avalanche stops when its momentum is lower than a user-defined fraction of its maximum momentum. Nevertheless, this
criterion is nonphysically based, because it depends on the avalanche's characteristics at a very different location and time.
Additionally, Bartelt et al. (2015) proposed the inclusion of an additional friction term related to snow cohesion, a real
physical snow property, which has an effect of retention and can stop the avalanche irrespective of the maximum momentum
reached during the avalanche propagation.

Herein, the friction–cohesion model (Voellmy–Salm plus cohesion), was analysed as part of the SWE terms of shear stress
along with its effect on mass and momentum equations. The relevance of each term was tested by comparing the numerical
results with well-documented laboratory experiments and with a real case study. Simulations were performed using the
numerical tool Iber (Bladé et al., 2014a), which is a two-dimensional (2D) hydraulic model; Iber has been recently enhanced
to simulate snow avalanches (Torralba et al., 2017). To that end, a specific numerical treatment of the friction–cohesion
model was implemented to adapt it to the particularities of the numerical scheme used by Iber (the Godunov method together
with the Roe Approximate Riemann Solver). The discussions on these numerical implementations, together with some other
considerations like the usage of nonhydrostatic pressure or nonisotropic properties, indicate that there is still a strong need
for research on the description and modelling of the whole avalanche process (triggering, release, motion, detention, etc.).



## 2 Materials and methods

### 2.1. 2D numerical modelling of non-Newtonian shallow flows

Most existing avalanche simulation models are based on the solution of mass and momentum conservation equations, which are similar to the equations for free surface water flows and differ only in the terms describing friction (rheological model). These equations, when applied to water, are named 2D Saint Venant equations, or 2D depth-averaged SWE (2D-SWE). They are derived from Navier–Stokes equations through a time average to filtrate the turbulent fluctuations (Reynolds equations, RANS) and a depth average to convert the 3D equations into 2D (Tan, 1992; Toro, 2009).

2D-SWE are a hyperbolic nonlinear differential system of three equations in partial derivatives, which, when expressed in compact vectorial notation, result in the following:

$$\frac{\delta}{\delta t} U + \nabla F = H \tag{1}$$

where $U$ is the temporal variation of the conservative variables, $F$ is the flow tensor, and $H$ is the source term. Momentum equations contain the gradients of the pressure and inertia terms (through the flow tensor $F$), the bottom slope and friction terms (through the source term $H$).

Using 2D-SWE-based numerical models to simulate non-Newtonian flows, such as snow avalanches, requires two additional hypotheses: *a monophasic fluid*, wherein the fluid is formed by a unique phase where all component are perfectly mixed, and *shear stress grouping*, wherein the shear stress can be considered as the sum of five different components (Julien and León, 2000) as follows:

$$\tau = \tau_d + \tau_t + \tau_v + \tau_{mc} + \tau_c \tag{2}$$

where $\tau_d$ is the dispersive term, $\tau_t$ is the turbulent term, $\tau_v$ is the viscous term, $\tau_{mc}$ is the Mohr–Coulomb terms, and $\tau_c$ is the cohesive term.

In general, this shear stress description can be used for the numerical modelling of non-Newtonian flows using the appropriate rheological model (Scheidl et al., 2013). Thus, if for water flow the shear terms due to friction are expressed by means of the friction slope, which is part of the source term $H$, then the rheological model for non-Newtonian fluids, which is expressed by the $S_{rh}$ term, can also be considered as a friction slope caused by the aforementioned components of the total shear stress.

In addition, non-Newtonian flows may have nonisotropic properties and nonhydrostatic pressure distribution, and new material may be entrained into the flow mass along its path. Nonhydrostatic pressure distribution can be accounted for through the coefficient $K_p$, which affects the pressure terms in the flow tensor $F$. Entrainment ($E$), or incorporation of new material, can be added in the mass conservation component of the source term $H$. Thus, Eq. (1), when applied to non-Newtonian flows, particularly to snow flows, can be written as follows:



$$U = \begin{bmatrix} h \\ hv_x \\ hv_y \end{bmatrix}; \quad F = \begin{bmatrix} hv_x & hv_y \\ hv_x^2 + K_p g \dfrac{h^2}{2} & hv_x v_y \\ hv_x v_y & hv_y^2 + K_p g \dfrac{h^2}{2} \end{bmatrix}; \quad H = \begin{bmatrix} E \\ gh(S_{o,x} - S_{rh,x}) \\ gh(S_{o,y} - S_{rh,y}) \end{bmatrix} \tag{3}$$

where $h$ is the flow depth, $v_x$ and $v_y$ are the two velocity components, $g$ is the gravitational acceleration, $S_{o,x}$ and $S_{o,y}$ are the two bottom slope components, and $S_{rh,x}$ and $S_{rh,y}$ are the two components of the rheological model. For water flows, the $K_p$ factor is equal to 1 (hydrostatic pressure), and $E$ is the a variation rate of the fluid column at a specific point, for example, a source or a sink (Bladé et al., 2019b), or during rainfall/infiltration processes in hydrological modelling (Cea and Bladé, 2015).

In particular, the friction terms $S_{rh}$ can be split into two terms [Eq. (4)]: $S'_{rh}$ for the flow resistance forces [Eq. (5)]; and $S''_{rh}$ for the cohesion forces [Eq. (6)]. The Voellmy–Salm friction model, which is widely used for granular flows such as snow avalanches, integrates the total flow resistance as the sum of a solid phase ($\mu$) and a turbulent resistance ($\xi$). The cohesion forces, added by Bartelt et al. (2015), can be defined as an additional flow resistance that depends on the cohesion ($C$) and normal stress. All terms are expressed as a friction slope as follows:

$$S_{rh} = S'_{rh} + S''_{rh} \tag{4}$$

$$S'_{rh} = \mu + \frac{v^2}{\xi h} \tag{5}$$

$$S''_{rh} = \frac{1}{\rho gh} C(1 - \mu)\left(1 - e^{-\frac{\rho gh}{C}}\right) \tag{6}$$

where $\rho$ is the flow density, $\xi$ is the turbulent friction coefficient, $\mu$ is the Coulomb friction coefficient, and $C$ is the cohesion parameter.

### 2.2. Numerical model

The previous equations have been implemented in Iber (Bladé et al., 2014a), a 2D numerical tool for simulating shallow flows in rivers and estuaries (www.iberaula.com). Initially developed for hydrodynamic and sediment transport simulations (Bladé et al., 2014b, 2019a), Iber has been continuously enhanced including different modules, such as hydrological processes (Cea and Bladé, 2015), water quality (Cea et al., 2016), large-wood transport (Ruiz-Villanueva et al., 2014), physical habitat suitability (Sanz-Ramos et al., 2019a), and, recently, non-Newtonian flows such as wood-laden flows (Ruiz-

Villanueva et al., 2019) and snow avalanches (Sanz-Ramos et al., 2020; Torralba et al., 2017).

Iber solves the described 2D-SWE through a conservative finite volumes scheme and on unstructured meshes of triangles and quadrilaterals. It uses a first-order Godunov-type upwind scheme for convective fluxes and the geometric slope source term, in particular the Roe scheme (Roe, 1986), and a centred scheme for the turbulent diffusion friction source term (Bladé




et al., 2014a). Therefore, the scheme achieves balancing of the bottom slope source term with the flow tensor, thereby

avoiding spurious oscillations of the free surface and retaining quiescent water even when working with complex irregular geometries (Bermúdez et al., 1998).

The model uses different numerical approaches for the spatial discretization with the finite volume method (Tan, 1992; Toro, 2009). Briefly, centred schemes use a linear interpolation without considering the flow direction, whereas upwind schemes are based on the characteristics theory for hyperbolic systems (LeVeque, 2002; Vázquez-Cendón, 1999) and consider the

fact that flow perturbations are propagated along the characteristic lines in space–time. Thus, upwind schemes consider the flow velocity and propagation direction. Schematically, the average values of the flow variables (depth and velocity) are stored at the geometric centre of the finite volume, but they are updated with the flows (mass flow or discharge and momentum flow) through the finite volume edges, which are computed with a noncentred stencil and the source terms. Therefore, noncentred discretization for the slope source terms ($S_o$) should be used. In other words, the slope is discretized

by a series of "steps" ($\Delta z$) between horizontal surfaces (finite volumes) instead of a continuous sloping surface.

Stopping of an avalanche on a sloping terrain is due to the equilibrium between friction terms and gravity forces at that time. Thus, when implementing the friction–cohesion model into the numerical scheme, a proper balance between the Coulomb stresses ($\mu$) and the cohesion stresses ($C$) must be ensured, which explain why these two frictions terms are treated separately from the turbulent stresses ($\xi$) in the numerical scheme.

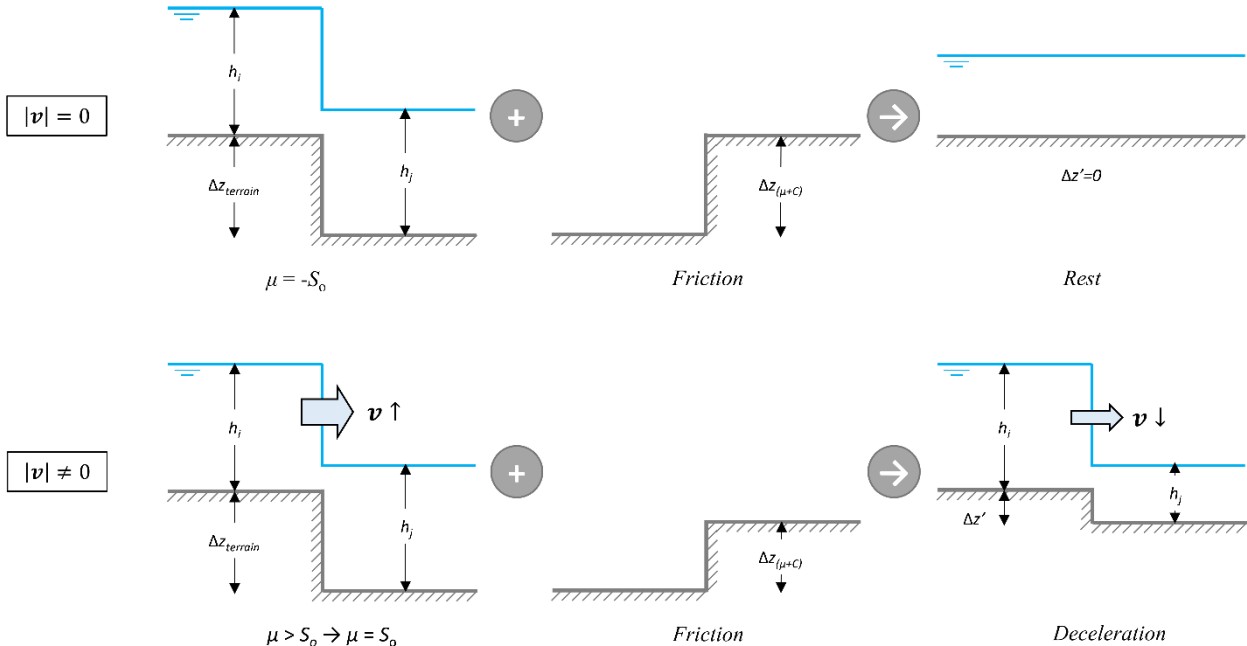

**Figure 1. Sketch of the numerical treatment of Coulomb ($\mu$) friction stresses, which can conceptually be indicated as a "friction step". The upper part of the figure illustrates how the geometric step ($\Delta z$) (upper left) is counterbalanced by the "friction step" ($\Delta z_{(\mu+C)}$) (upper middle), and thus the velocity is kept null (upper right). The lower part shows how the friction step (lower middle) opposes the gravity forces, and thus the calculated velocity (lower figure right) is less than the velocity in the case of no friction (lower figure left).**





Coulomb solid friction and cohesion ($C$) terms are discretized by an upwind numerical scheme, similar to the scheme used for the bottom slope. Conceptually, this could be interpreted as the addition of an elevation difference to the mentioned geometric bottom "step" across finite volume edges, which can be indicated as a "friction step" ($\Delta z_{(\mu+C)}$) against flow motion (Figure 1).

Turbulent friction is treated through a centred scheme following the same methodology as for water flow, with the difference

being that, for water, the friction slope is calculated using the Manning equation, whereas, the turbulent part of the Voellmy–Salm model equation is employed herein. A detailed description of the numerical schemes implemented in Iber for water flow can be found in the research by Bladé and Gómez-Valentín (2006), and the treatment of the source term is described in detail in Brufau et al. (2002) and Bladé et al. (2012).

### 2.3. Model analysis and description of the case studies

To validate the numerical model, the formulation used for the shear stress terms of the Voellmy–Salm and cohesion models was first analysed. This analysis focuses on understanding the behaviour of avalanches by varying the involved variables ($\mu$, $\xi$, $C$, $h$, $v$ and $\rho$) within the common ranges described in the literature, and exploiting their contribution to the total shear stress ($\tau = \tau_\mu + \tau_\xi + \tau_C$). The second part of the validation includes the simulation of two well-documented experimental case studies. The first experiment is a laboratory flume with granular material (glass beads) (Hutter et al., 1995), whereas the

second one is a "natural" channel with snow flow (Lang and Dent, 1980). These tests aimed to calibrate the parameters of the friction–cohesion model in the numerical modelling to match the results with the experiments. Finally, a recorded snow avalanche that took place in the Pyrenees in 2018 was simulated in order to test the numerical model in real conditions. Different parameter combinations of the friction–cohesion model were analysed to identify the numerical model response to the parameter variations in terms of runout distance and cumulated snow.

### 2.3.1. Case 1 (Hutter, Exp. 117)

Hutter et al. (1995) performed laboratory experiments to study the flow motion of granular flows, such as landslides, rockfalls and ice, and dense flow snow avalanches, under controlled conditions. The experiments consisted of a chute of two straight channels 0.1 m wide with different slopes joined by a circular transition zone. On the upper part, the granular material was stored and then released with almost instantaneous gate opening.

The experiment (Exp. 117) consisted of a 1.4 m long channel, inclined by 60º, and 1.7 m long horizontal part connected by a 0.258 m long curved part. From the storage area, 4 kg of glass beads (1,730 kg m$^{-3}$) was released and then analysed with high-speed photography to determine the position of the avalanche.

The chute was discretized using distorted rectangles elements (0.01 m in perpendicular direction and 0.1 m in the direction parallel to the flow), because the flow is mainly 1D. A wet–dry limit of 0.003 m, which is equal to the material size, was

used. The simulation time was 2 s with intervals of 0.05 s, while the initial conditions included the material initially





positioned in the storage area and dry conditions in the rest. Only the friction terms $\mu$ and $\xi$ were analyzed. No cohesion was considered owing to the material's nature.

### 2.3.2. Case 2 (Lang and Dent, 1980)

Lang and Dent (1980) built up a seminatural facility to obtain and compare data with the results of a computer model. This facility was constructed near a ski resort and consisted of a semicircular inclined (30º) channel lined with a plastic sheet aiming to achieve maximum flow velocities up to 18 m s⁻¹. A 2.4 m wide horizontal deceleration area existed immediately after the inclined part. The position of the leading-edge was filmed and transformed into a function of time.

Herein, Experiments 1 and 3 (Lang and Dent, 1980), denoted as Exp. 1 and Exp. 3 respectively, associated with terminal velocities of 12 and 18 m s⁻¹, respectively, were analysed. The setup of the numerical model consisted only of the flat part of the chute and was discretized by 1D elements of 0.01 m. An initial snow volume of 3.36 m³ was introduced in the model with the terminal velocity as an initial condition. Since no data were available, the flow density was assumed to be 300 kg m⁻³. The simulation duration was 3 s with a time step of 0.1 s and the wet–dry limit was 0.01 m. The influence of the three parameters, namely $\mu$, $\xi$, and $C$, was studied.

### 2.3.3. Case 3 (Coll de Pal, 2018)

Coll de Pal is a mountain pass (2,070 m.a.s.l.) located in the central part of the Catalan pre-Pyrenees range, northeast Spain (Figure 2a). A road (BV-4024) that crosses the pass, is exposed to snow avalanches during the winter season.

On February 10, 2018, a slab avalanche event occurred at the area known as Rocs de la Bòfia between the 17ᵗʰ and 18ᵗʰ km of the road. The avalanche flowed from an unknown triggering area and stopped several meters below the road. Two days after the event, the runout area and snow depths on the road were recorded (Figure 2b, coloured in blue).

The study area is within the zone called RIT051 according to Avalanche Database of Catalonia (BDAC) and contains the potential triggering and runout area of the avalanche (Figure 2b). The release area of this event was estimated by means of i) a field recognition, mostly tacking into account roughness and damage evidence on vegetation; ii) a terrain slope analysis considering that slopes below 28 º and above 45º were not adequate to generate an avalanche (stable condition) and to keep enough snow thickness to trigger an avalanche, respectively; and iii) a slope side orientation analysis considering the preferential snowdrift accumulation area due to wind action previous to the event. The result of this analysis is displayed in yellow in Figure 2b.

The study area was discretized in the numerical model through an unstructured mesh of triangles with a side length of 2 m. An initial condition of 0.80 m of snow was imposed on the selected release area (Figure 2b, coloured in yellow). The flow density was assumed to be 300 kg m⁻³ (no data were available), and a wet–dry limit of 0.01 m was chosen. Different combinations of the friction–cohesion model parameters were tested for a simulation of 180 s. Results of the runout distance from the lower part of the release area to the furthest position were compared. In addition, the snow depth at the end of the event and the maximum velocity achieved during the flow motion were analysed.


Figure 2. Coll de Pal case study: (a) General location (red point) of the case study Coll de Pal (background image source: ESRI). (b) Representation of the RIT051 zone (purple), the selected release area (yellow) and the recorded runout area of the event (blue) (background image source: Institut Cartogràfic i Geològic de Catalunya).

## 3. Results

### 3.1. Formulation analysis

The parameters of the Voellmy model $(\mu, \xi)$ and cohesion $(C)$ were analysed individually, focusing on their significance from the viewpoint of shear stress. This review was performed for a constant value of density of 300 kg m$^{-3}$, which can be associated with relatively dense snow in natural state. Density is involved in all terms as a proportionality factor; however, if the friction terms are expressed as a friction slope, as in Eq. (5) and Eq. (6), the density does not appear explicitly, except for the cohesion in which it acts as a reduction factor $(\rho^{-1})$. The friction slope is proportional to the shear stress through a factor of $\rho g h$; thus, for shear stresses density is relevant.

### 3.1.1. Coulomb friction stress ($\tau_\mu$)

Coulomb friction stress ($\tau_\mu$) is linearly dependent on the flow depth ($h$) and the dry-Coulomb friction coefficient ($\mu$) [Eq. (5)]. Thus, if it is calculated by varying these values, a cone-shaped surface will be generated (Figure 3a), where $\tau_\mu$ values are plotted with intervals of 1,000 Pa when $h$ varies from 0 to 2.5 m and $\mu$ from 0 to 1, resulting in $\tau_\mu$ values lower than 7,500 Pa (for a density of 300 kg m⁻³).

However, values of $\mu$ lower than 0.1 or greater than 0.6 might be unrealistic (Platzer et al., 2007a, 2007b). Therefore, the

shear stress due to the Coulomb friction is not supposed to exceed 4,500 Pa for snow in the natural state, when density is near 300 kg m⁻³. For other density values, this limit can increase to 11,000 Pa, e.g., for slush snow, which can reach a density of up to 750 kg m⁻³ (Jaedicke et al., 2008; Platzer et al., 2007a).

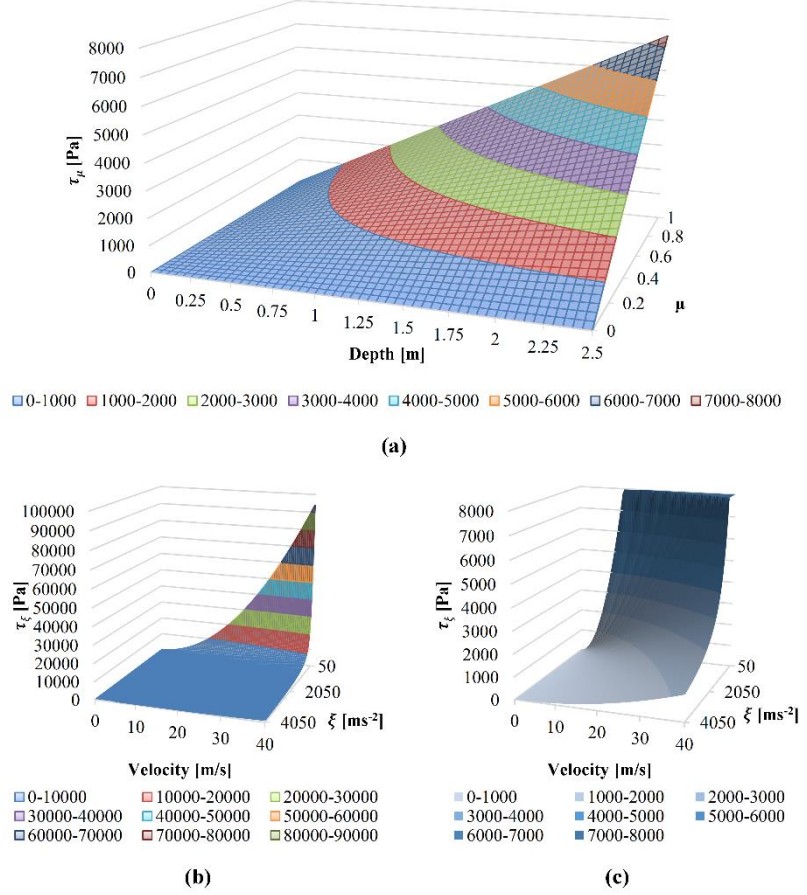

**Figure 3. Shear stress contribution due to Voellmy-Salm model: (a) Coulomb friction ($\tau_\mu$, in Pa) as a function of the flow depth ($h$) and the Coulomb friction coefficient ($\mu$); (b) Turbulent friction ($\tau_\xi$, in Pa) as a function of the flow velocity ($v$) and the turbulent**

**friction coefficient ($\xi$); (c) A zoomed-in view of the blue region of (b), where in $\tau_\xi$ is limited to 8,000 Pa.**

Natural Hazards and Earth System Sciences
Author(s) 2020




### 3.1.2. Turbulent friction stress ($\tau_\xi$)

As mentioned before, the turbulent friction stress ($\tau_\xi$) depends on the square of the velocity ($v$) and is inversely proportional to the turbulent friction coefficient ($\xi$) [Eq. (5)]. This term was evaluated in this study by varying both $\xi$ and $v$. Figure 3b shows the variation range of $\tau_\xi$ when $v = 0$–$40$ m s$^{-1}$ and $\xi = 50$–$4,050$ m s$^{-2}$. Figure 3c illustrates an enlargement for $\tau_\xi$

values less than 8,000 Pa.

Low velocities ($< 10$ m s$^{-1}$) and moderate values of $\xi$ ($> 900$ m s$^{-2}$) resulted in a limited contribution of the frictional terms ($< 300$ Pa). For this range of velocity and very low $\xi$ values ($> 450$ m s$^{-2}$), the shear stress was lower than 1,700 Pa. Stress values above 10,000 Pa require $\xi < 450$ m s$^{-2}$ and velocities greater than 13 m s$^{-1}$.

The effect of density on $\tau_\xi$ was proven to be similar to that on $\tau_\mu$. Consequently, different snow density values would lead to

different shear stress behaviours. For example, slush snow would result in double shear stress values for the same values of $v$ and $\xi$.

### 3.1.3. Cohesion friction stress ($\tau_C$)

The cohesion contribution to the shear stress ($\tau_C$) depends on $\mu$ and on the cohesion parameter $C$. Equation (6) can be split into two terms: $C(1 - \mu)$ and $\left(1 - e^{-\frac{\rho g h}{C}}\right)$. The first is linearly dependent on $C$, and on $\mu$ in through the term $(1 - \mu)$, a

reduction term. According to Platzer et al. (2007a, 2007b), since $\mu$ is between 0.1 and 0.6, the reduction term $(1 - \mu)$ is between 0.4 and 0.9. The second term depends on the depth ($h$), density ($\rho$) and cohesion ($C$).

Shear stress was evaluated for a range of $C$ between 50 and 2,050 Pa, as suggested by Bartelt et al. (2015), whereas $h$ ranged from 0 to 5 m and $\mu$ from 0 to 0.5. Figure 4 shows $\tau_C$ for $C = 250$ Pa (Figure 4a) and $C = 2,000$ Pa (Figure 4b). An accretion of $\tau_C$ with $h$ can be observed, more accentuated for lower values of $h$; and a linear diminution of the shear stress, regardless

of the flow depth, while $\mu$ increases.

Figure 4c exhibits the performance of the aforementioned second term. This term is 0 when $h$ tends to 0 or $C$ tends to $\infty$, and it is 1 for high values of $h$ or for $C$ tending to 0. The effect of this term is quite limited on $\tau_C$. Mid-low $\tau_C$ values (0–0.75) are obtained for flow depths lower than 0.9 m with a flow density of 300 kg m$^{-3}$. Thus, $\tau_C$ is an important parameter only for low values of $h$, affecting the spreading areas (lateral and tail).


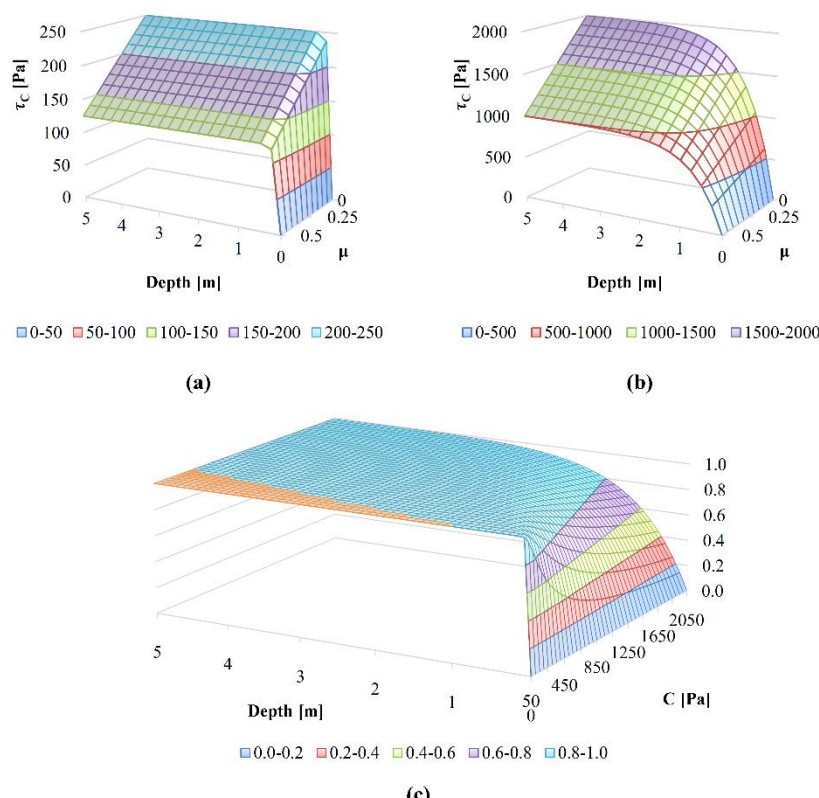

**Figure 4.** Shear stress contribution due to cohesion: (a) Cohesion friction ($\tau_C$, in Pa) for $C = 250$ Pa as a function of the flow depth ($h$) and the Coulomb friction coefficient ($\mu$); (b) Cohesion friction ($\tau_C$, in Pa) for $C = 2,000$ Pa as a function of the flow depth ($h$) and the Coulomb friction coefficient ($\mu$); (c) Representation of the term $\left(1 - e^{-\frac{\rho g h}{C}}\right)$ as a function of the flow depth ($h$) and the cohesion ($C$).

## 245  3.2. Case 1 (Hutter, Exp. 117)

This case, presented by Hutter et al. (1995), was used to study the effect of the parameters $\mu$ and $\xi$ on the dynamics of granular flows. The cohesion effects were not considered because of the material's properties. Numerical results were compared with the observations in terms of the rear (r) and front (f) position of the avalanche. In the following paragraphs and figures, $\mu X\_\xi Y$ notation is used to refer to simulation with values X and Y for $\mu$ and $\xi$, respectively. The parameter $\mu$

varies from 0.19 to 0.49 and $\xi$ varies from 500 to 2,000 m s$^{-2}$. The upper limits were chosen according to Bartelt et al. (1999). The results were also compared with the simulations presented in the same study.

Figure 5 illustrates the results of Exp. 117 for $\mu = 0.49$ and different values of $\xi$. Good fitting at the first time steps (t < 0.3 s) is observed, but there is an overestimation during the displacement of the material and less spread of the avalanche. Using the parameters proposed by Bartelt et al. (1999), a good adjustment is observed for the end position of the avalanche front,

but the rear position of the avalanche is overestimated (approximately 0.45 m). In contrast, a good adjustment of the rear position of the avalanche is achieved with $\xi = 1,000$ m s$^{-2}$, but then the front position is underestimated (around 0.35 m).
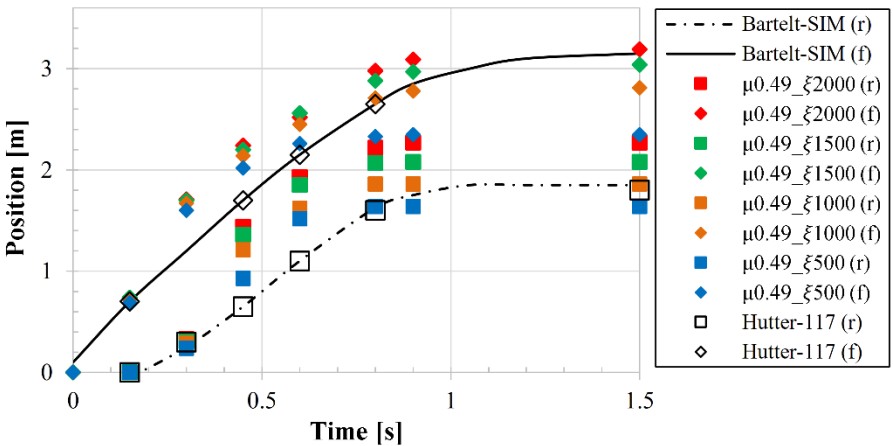

**Figure 5. Comparison between the measured positions (r: rear; f: front) by Hutter et al. (1995) and the computed results using different combinations of *μ* and *ξ* during Exp. 117. The lines represent the results of the computed simulation by Bartelt et al. (1999).**

Different $\xi$ values (0.19, 0.29, and 0.29 m s$^{-2}$) were also tested. In general, the same patterns can be observed (Figure 6). After $t = 0.3$ s, the velocity increases, resulting in a larger rear and front positions and further expansion of the avalanche. Small differences can be identified on the simulated rear part of the avalanche, whereas the runout of the front part decreased when $\mu$ and $\xi$ increased.

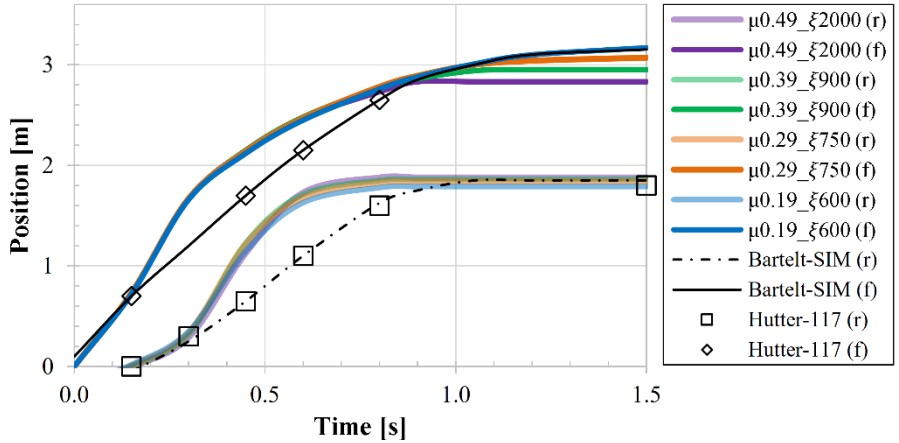

**Figure 6. Best numerical results of the rear and front positions of the avalanche compared to the measured data (experiment**
**no. 117 performed by Hutter et al., 1995). The black lines represent the results of the computed simulation by Bartelt et al. (1999).**

The two numerical models (Bartelt et al., 1999, and Iber) exhibited different behaviours. In terms of terminal velocity, Iber generally achieves values up to 0.5 m s$^{-1}$ higher, indicating larger inertial forces, thereby reducing the avalanche spread. The discrepancies in the numerical results can be attributed to (i) the type of the numerical model used (Bartelt et al., 1999, used a 2D model in the vertical); (ii) the different numerical scheme (Bartelt et al., 1999, used a curvilinear formulation and a





finite differences scheme); and (iii) the use of a constant velocity as the initial condition (the avalanche release was
simulated with Iber, no-initial velocity was applied).

### 3.3. Case 2 (Lang and Dent, 1980)

Experiments 1 and 3 of Lang and Dent (1980) were used to analyse the flow behaviour on the snow avalanche propagation,
from the terminal velocity (the snow enters the decelerating zone) to the stop point. Several combinations of the parameters

$\mu$ (0.1–0.3), $\xi$ (5,500–10,000 m s$^{-2}$) and $C$ (490–1,060 Pa) were tested.

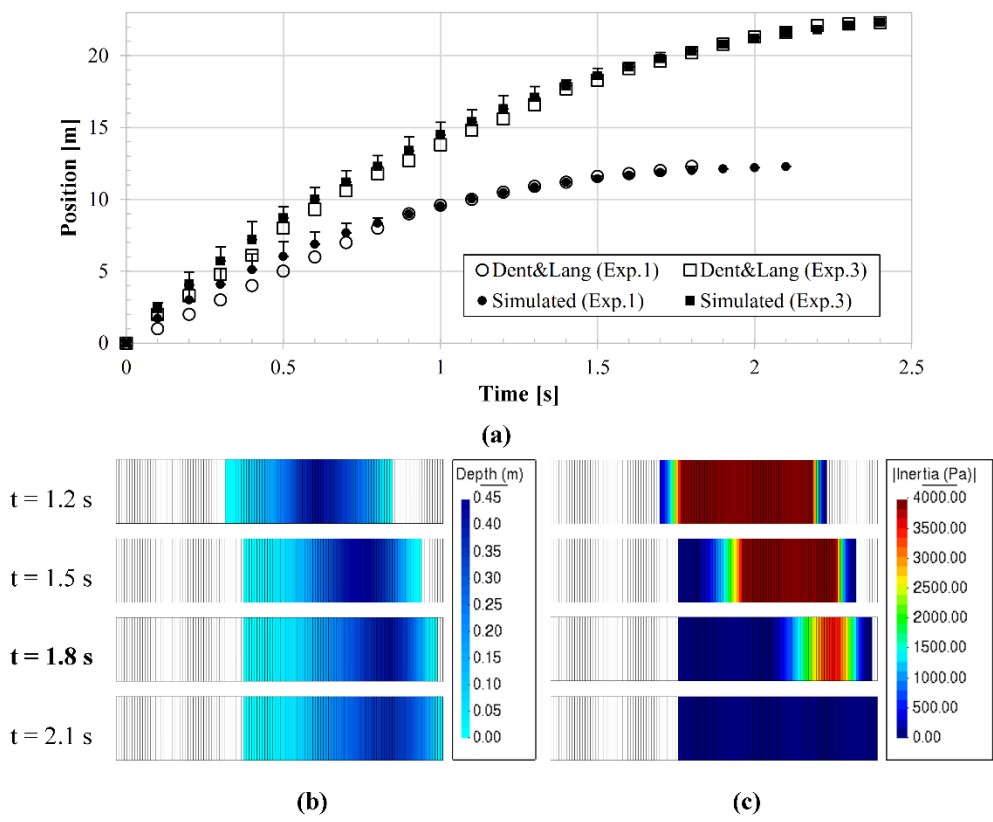

**Figure 7. Results of the simulation of the experiments performed by Lang and Dent (1980): (a) Leading-edge position of all simulation (mean values) versus time (black) compared with Exp. 1 and Exp. 3 (white); Evolution of the flow depth (b) and inertial forces (c) for the Exp. 1 (values above 4,000 Pa are coloured in deep red; view of the first 12.3 m of the channel).**

Figure 7a shows the average value for all the simulations (black) of the leading-edge position versus time in comparison with

the reported data (white) for Exp. 1 and Exp. 3. In general, an overestimation of the avalanche front at starting times is
observed, but the final position at the end of each experiment (1.8 s for Exp. 1 and 2.4 s for Exp. 3) is well captured for all
simulations. More significant differences can be identified for Exp. 1 than for Exp. 3. A possible explanation is the spreading
of the leading-edge position (in the experiments, the snow flows as a block) and the initial condition in the numerical model
that was according to Lang and Dent (1980) (analysed in deep in the *Discussion section*). In addition, the simulations of

Exp. 1 require 0.3 s more for the avalanche to stop, resulting in a run-over approximately 0.25 m larger. The analysis of the




inertial forces can also help understand the evolution of the avalanche shape. In Figure 7b, it can be seen that the simulated avalanche stops at the rear part, but the inertial terms keep pushing the central and front-parts of the avalanche (Figure 7c). This model configuration, with a constant velocity as initial boundary condition, leads to higher inertial terms on the rear avalanche part in initial time steps, but then the inertia trespasses from the rear to the front part, and in concordance decreases whereas the depth increases.

A good adjustment can be obtained for a wide combination of values of the three analysed parameters. Different combinations were used for both experiments, showing the possibilities provided by these parameters for model calibration. In comparison with the previous test cases, high $\xi$ values (up to 10,000 m s$^{-2}$) are required to achieve good results. These values are two or three times greater than those more commonly found in the bibliography, but in line with the values obtained by Fischer et al. (2015).

Figure 8a and Figure 8b depict the surface generated by the combination of $\mu$, $\xi$ and $C$ that better approximates the observed results of the final leading-edge position. Based on these results, a linear dependency for cohesion ($R^2 = 0.97$) was obtained for Exp. 1 (Figure 8a), in which the cohesion is proportional to the turbulent friction coefficient and the Coulomb coefficient, as shown in Eq. (7). Furthermore, in case of Exp. 3 (Figure 8b), cohesion is log-dependent on the Voellmy parameters, as shown in Eq. (8), with $R^2$ greater than 0.99.

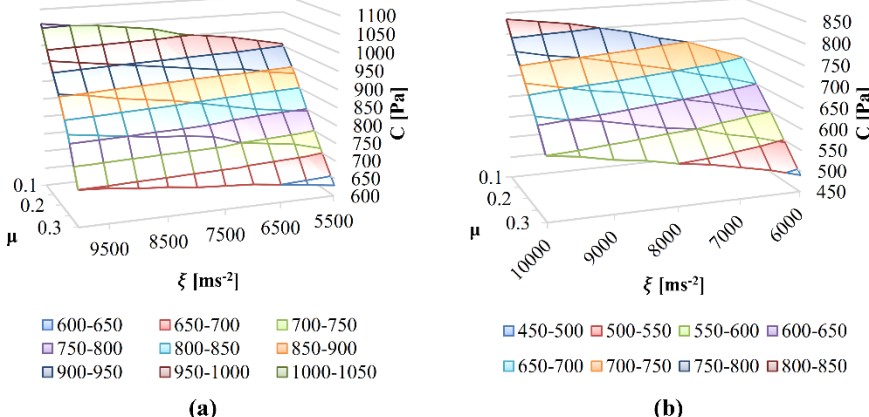

**Figure 8. Representation of the best fit surfaces generated by the combination of $\mu$, $\xi$ and, $C$ that better approximates the observed results of the final leading-edge position: (a) Exp. 1; (b) Exp. 3.**

The determination of cohesion for Exp. 1 and Exp. 3 resulted in Equations (7) and (8), with a good adjustment even for values that were out of the already reported range. However, the validity of these expressions is limited to $\mu < 0.7$ for Eq. (7) and for $\mu = \xi = 0$ for Eq. (8).

$$C = \alpha \cdot \xi + \beta$$
$$\alpha = -0.025 \cdot \mu + 0.0265$$
$$\beta = -1,506.4 \cdot \mu + 980.66$$

(7)



$$C = \gamma \cdot ln(\xi) + \delta$$
$$\gamma = -255.32 \cdot \mu + 289 \tag{8}$$
$$\delta = -216 \cdot ln(\mu) + 1,109.3$$

### 3.4. Case 3 (Coll de Pal, 2018)

The snow avalanche of February 10, 2018, which occurred in Coll de Pal, was simulated with different combinations of the friction–cohesion model parameters $\mu$, $\xi$, and $C$. Using a uniform estimation of each parameter throughout the model, 27 scenarios were simulated (Table 1).

The recorded runout area was approximately 500 m in length and had a maximum width of 60 m few meters below the road. The maximum snow depth, located on the road at the west side of the avalanche, was of 2.4 m and gradually decreased to 1.3 m on the east side. Along the avalanche path, there is a gully where the maximum velocity was probably reached, as the maximum slopes were observed and there the flow had to narrow in order to pass through.

**Table 1. Combinations of the friction–cohesion model parameters $\mu$, $\xi$ and, $C$ performed in the Coll de Pal case study.**

| Combinations | A | B | C |
| --- | --- | --- | --- |
| | $\mu$ [-] | $\xi$ [m s$^{-2}$] | $C$ [Pa] |
| 1 | 0.2 | 250 | 0 |
| 2 | 0.4 | 1,000 | 100 |
| 3 | 0.6 | 2,000 | 500 |


Sensitivity analysis results are listed in Table 2. Increments in $\mu$ or $C$, or a reduction in $\xi$ result in larger friction stresses, shorter runout distances, higher avalanche depths, and a reduction of the maximum velocity. Parameter $\mu$ significantly affects the runout distance and snow depth accumulated on the road, but a lower effect is observed on the maximum velocity. In contrast, an increment in the value of $\xi$ results in lower friction and higher inertia, leading to larger avalanche travel

distances. The parameter $\xi$ also affects the avalanche dynamics, as inertia terms push the avalanche in the flow direction irrespective of the ground slope direction. Cohesion also play an important role in the snow avalanche tail definition as it affects the stopping moment of the avalanche, when its thickness diminishes.

The first scenario, A1B1C2 (Figure 9a), adjusts well the avalanche direction owing to the low values of $\xi$, but overestimates the accumulated snow on the road. Below the road, the snow avalanche flow continues and separates into three branches. A

long tail is then observed because of cohesion (100 Pa). Scenario A1B3C3 (Figure 9b) shows a good performance in terms of runout distance. However, the stopping area is shifted to the east (approximately 15 m), denoting the effect of the increased inertia due to the higher $\xi$, which prevents the change of the flow direction after the gully. For scenario A2B2C1 (Figure 9c), no final tail can be observed above the road there is no cohesion ($C = 0$ Pa). However, despite being no cohesion, some snow still remains in the release area, because in some sectors the slope is lower than $\mu$ (0.4). Finally,





scenario A2B3C2 (Figure 9d) is in very good agreement with the observations, in terms of both the runout distance and the snow depth on the road. Nevertheless, the simulated flow direction is again shifted because the inertia of the avalanche after leaving the gully area is maintained because of the high $\xi$ values.

**Table 2. Results of the simulated scenarios of the Coll de Pal case study. The scenarios highlighted in grey were analysed in detail. Each scenario is referred as AaBbCc where a, b and c are 1, 2 and 3, respectively (see Table 1).**

| Scenario | Runout distance [m] | Max. Road depth [m] | Max. velocity [m s⁻¹] | Observations |
|---|---|---|---|---|
| A1B1C1 | >475 | 2.3 | 18.1 | Continues flowing |
| A1B1C2 | 373 | 2.8 | 17.7 | |
| A1B1C3 | 266 | 3.4 | 16.4 | Stops on the road |
| A1B2C1 | >475 | 1.4 | 27.8 | Continues flowing |
| A1B2C2 | >475 | 1.8 | 27.3 | Continues flowing |
| A1B2C3 | 337 | 2.7 | 27.7 | |
| A1B3C1 | >475 | 0.8 | 35.0 | Continues flowing |
| A1B3C2 | >475 | 1.4 | 34.6 | Continues flowing |
| A1B3C3 | 390 | 2.1 | 37.7 | |
| A2B1C1 | 266 | 4.5 | 16.0 | Stops at the road |
| A2B1C2 | 266 | 4.4 | 15.7 | Stops at the road |
| A2B1C3 | 266 | 3.7 | 14.4 | Stops at the road |
| A2B2C1 | 370 | 3.3 | 24.3 | |
| A2B2C2 | 330 | 3.5 | 24.1 | |
| A2B2C3 | 266 | 4.0 | 26.4 | Stops at the road |
| A2B3C1 | 430 | 2.2 | 32.1 | |
| A2B3C2 | 385 | 2.6 | 32.4 | |
| A2B3C3 | 285 | 3.4 | 33.4 | |
| A3B1C1 | 263 | 4.5 | 15.0 | Stops at the road |
| A3B1C2 | 263 | 4.1 | 14.2 | Stops at the road |
| A3B1C3 | 258 | 2.2 | 13.5 | Stops before the road |
| A3B2C1 | 263 | 4.7 | 20.5 | Stops at the road |
| A3B2C2 | 263 | 4.2 | 20.7 | Stops at the road |
| A3B2C3 | 258 | 2.7 | 21.6 | Stops before the road |
| A3B3C1 | 266 | 4.5 | 31.5 | Stops at the road |
| A3B3C2 | 266 | 4.2 | 32.1 | Stops at the road |
| A3B3C3 | 258 | 2.7 | 29.0 | Stops before the road |



In summary, the best results are obtained when $\mu$ varies from 0.2 to 0.4, $\xi$ is between 250 and 2,000 m s$^{-2}$, and $C$ is less than 500 Pa.

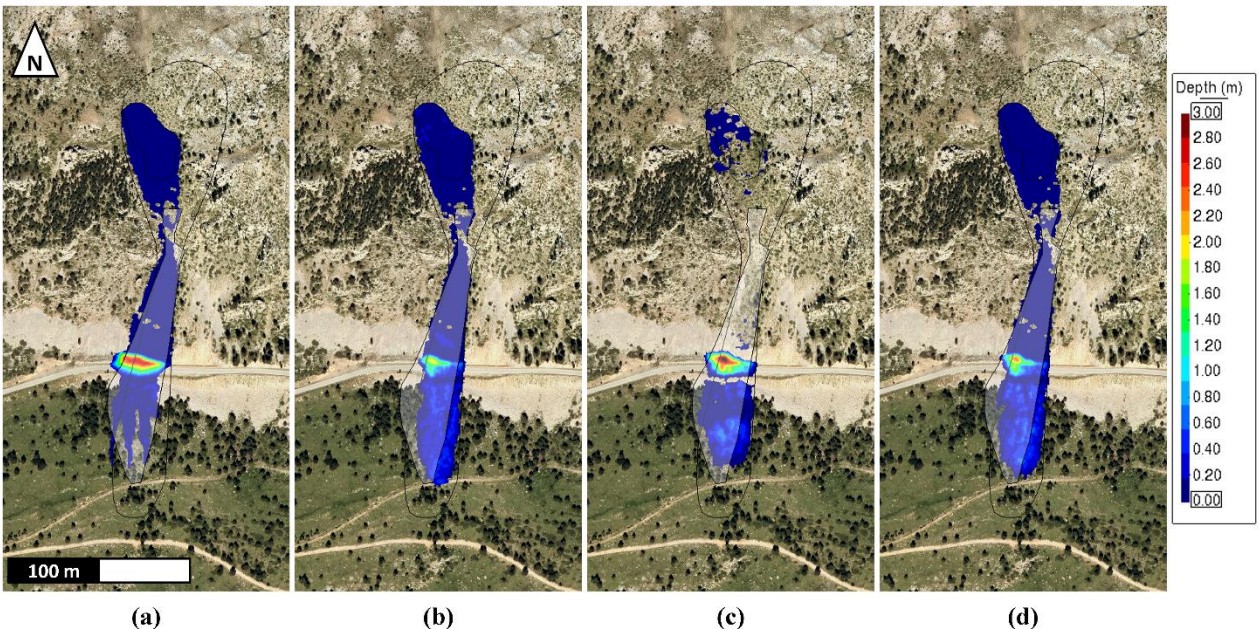

**Figure 9. Snow depth at the end of the simulation (180 s) for best fit scenarios: (a) A1B1C2; (b) A1B3C3; (c) A2B2C1; (d) A2B3C2. Dark line represents the region RIT051 (BDAC) and transparent-white region is the observed runout (background image source: Institut Cartogràfic i Geològic de Catalunya).**


Assuming a 100-year return period, the recommendations for parameter estimation by Bartelt et al. (2017) lead to a value of 0.34 for $\mu$, 1,250 m s$^{-2}$ for $\xi$, and 100 Pa for $C$. Figure 10a shows the slope vectors of the terrain and how their main directions are in concordance with the recorded avalanche.

The results of the simulation with the recommended parameters are in agreement with the observed data (Figure 10b). The

transition zone is within the observed perimeter, only small differences (< 5 m) are observed in the downstream half of this zone. The maximum velocities (approximately 25 m s$^{-1}$, Figure 10c) are reached in the gully area, a few meters downstream of the release area. The road also has a significant effect. Snow accumulates on the inner side, where a maximum depth of 2.5 m (west part) is achieved. Moreover, although the deposition shifts 15 m to the east, in accordance with the high value of the turbulent friction coefficient, the runout distance is well captured.

The identified differences can be attributed to the assumptions on the release area (shape, extension, and depth) and the use of summer topography, which can retain snow in some areas, and would probably be smoother in winter topography. Additionally, as previously observed, the avalanche would better follow the slope direction for lower values of the turbulent friction parameter ($\xi$ = 250 m s$^{-2}$).


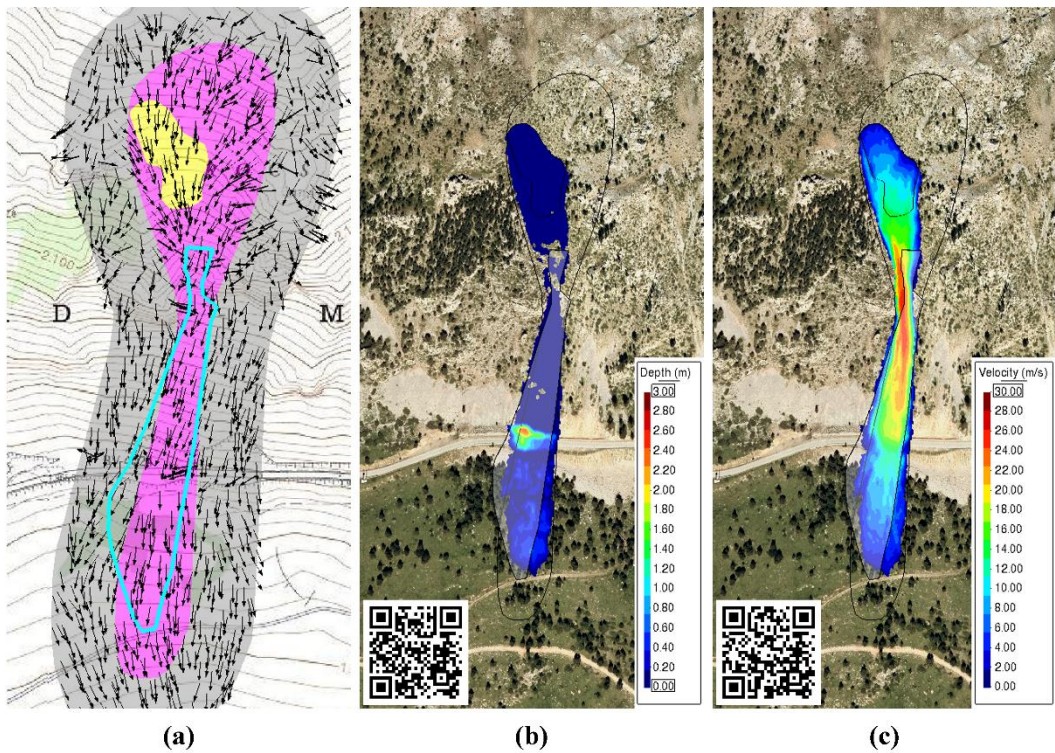

**(a)**        **(b)**        **(c)**

**Figure 10. (a) Representation of the main directions of the terrain slope on the study area. The grey region is the model domain, the purple region is the RIT051 area (BDAC), the yellow region is the release area, and the blue polygon is the observed avalanche perimeter. Results of the simulations for $\mu = 0.34$, $\xi = 1,250 \, \mathrm{m \, s^{-2}}$ and $C = 100 \, \mathrm{Pa}$ (scan the QR code to see the video): (b) Snow depth at the end of the simulation; (c) Map of maximum velocity achieved during the avalanche. In (b) and (c) the dark line represents the RIT051 region and the transparent-white region is the observed runout (background image source: Institut**
**Cartogràfic i Geològic de Catalunya).**

## 4. Discussion

### 4.1. Simulation of snow avalanche dynamics using 2D-SWE-based models

In the numerical modelling of avalanches with 2D-SWE-based models, flow is simulated as a continuum. Individual motion of particles that might happen in nature (Figure 11a) cannot be simulated with the applied methodology. However, as shown

in Figure 11b (video), most dense-snow avalanches behave as continuum and 2D-SWE can be used herein to describe the avalanche dynamics. The intrinsic hypothesis in 2D-SWE can produce uncertainties in the extension and internal movements of particles, but this happens for all numerical models, which are a simplification of the reality.

In the numerical modelling of flows it is necessary to establish a wet–dry limit depth, which is a threshold to consider whether there is flow in a mesh element or not. This is a relevant parameter for water flow, especially in flat areas (Cea et al.,

2007; Ramos-Fuertes et al., 2013; Sanz-Ramos et al., 2019b) and for hydrological modelling (Cea and Bladé, 2015; Sanz-Ramos et al., 2018b), but it also applies to non-Newtonian-fluid flows such as snow avalanches. Very large wet–dry limits,





greater than a few centimetres, can significantly alter the flow propagation, especially that of the flow front (Bladé et al., 2014a; Cea and Bladé, 2015). The wet–dry limit must be properly defined considering, in general, the geometric dimension of the problem, the mesh size, the expected flow depth and, in particular, the fluid properties. In addition, selecting the

appropriate numerical scheme to deal with wet–dry fronts, especially the drying method, is fundamental for preserving the mass conservation. A detailed analysis of the drying methods implemented into Iber can be found in previous studies (Bermúdez et al., 1998; LeVeque, 2002; Roe, 1986; Vázquez-Cendón, 1999). However, no references on wet-dry limit treatment could be found for dense-snow avalanche models based on the 2D-SWE. For this type of flow, the particle size of the snow aggregates is generally larger than a few centimetres; thus, wet–dry limit values of 1–5 cm are sufficient to

properly define the dynamics and the extension of the avalanche. However, the use of lower wet–dry limits should not be discarded.

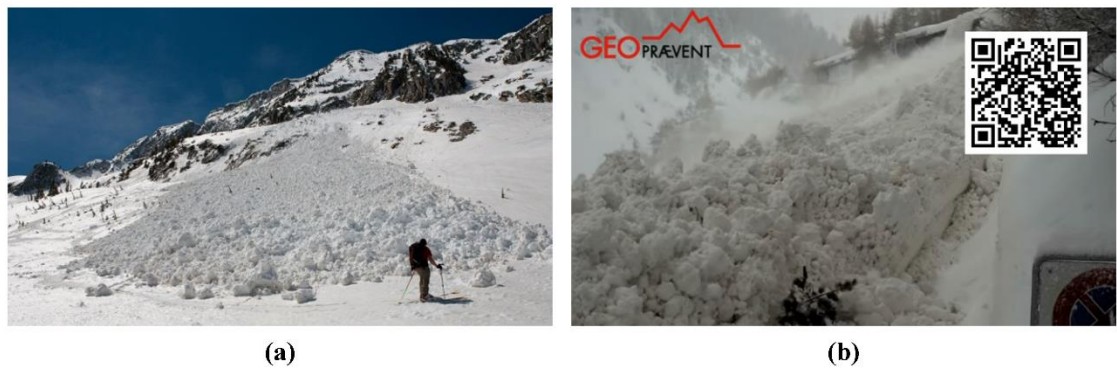

| (a) | (b) |

**Figure 11. Different avalanche types which are desegregated by particles: (a) Wet slide avalanche (Zimmerman-Wall and Burr, 2014); (b) Video (scan QR) of a gully avalanche registered between Zermatt and Täsch (GEOPRAEVENT, 2018).**

As previously indicated, when using the 2D-SWE to simulate non-Newtonian flow, like snow avalanches, the characteristics

of the fluid are mainly considered through the source term $H$ of Eq. (4), which represents the rheological model. However, the numerical treatment of this term depends on the numerical method (finite differences, finite elements, finite volumes, etc.) and the numerical scheme (centred, upwind). This treatment must ensure the balance between the source term ($H$) and the flow tensor ($F$) (Bladé and Gómez-Valentín, 2006). In the currently developed numerical model, based on Iber, which uses the Roe scheme, a centred scheme is used for the turbulent stress terms and a decentred scheme is used for solid friction

and cohesion.

**4.2. On nonhydrostatic anisotropic pressure distribution**

For water, the 2D-SWE usually assume a hydrostatic and isotropic pressure distribution (Chaudhry, 2008). This means a linear variation in the vertical direction with the specific weight of the flow and the same in all horizontal directions. However, for non-Newtonian flows and steep slopes, this premise cannot be realistic (Ruiz-Villanueva et al., 2019). In


particular, for snow avalanches, especially during the avalanche release, the assumption of anisotropic pressure distribution

can improve the definition of the avalanche dynamics.

Several reported studies have considered anisotropic pressure distribution (Bartelt et al., 1999; Hungr, 1995; Hungr and

McDougall, 2009; Ruiz-Villanueva et al., 2019). This correction is commonly made through a factor ($K_p$) that multiplies the

pressure terms in the momentum equations. Indeed, using a $K_p$ value equal to 1 when simulating avalanches (with the

Voellmy–Salm and cohesion model) leads to surface shapes similar to those of water flows. In order to illustrate this, a

dummy flat channel with a width of 5 m, a length 20 m, and an initial volume of 50 m$^3$ was simulated for water and snow

flows. In it, a dam-break like flow was simulated, instantaneously releasing the fluid from an area with an initial depth of

2 m at one end of the channel. Figure 12a shows the free surface evolution for water and snow flows with time increments of

0.5 s. For snow flow, only turbulent friction ($\xi = 1{,}600$ m s$^{-2}$) was implemented, with two different $K_p$ values (1 and 0.5).

The flow behaviour is almost identical for water (blue lines) and snow with $K_p = 1$ (green dashed lines). For $K_p = 0.5$ (brown

dotted lines), the flow is slower, but the free surface acquires similar shapes as for $K_p = 1$. The pivoting point of the free

surface is the same for all simulations, maintaining the length and depth positions in approximately 5 m and 0.9 m,

respectively. The inertia terms, shown in Figure 12b, are also very similar for water and for snow with $K_p = 1$, and are

approximately twice those for $K_p = 0.5$, highlighting the effect of $K_p$ on flow propagation. Indeed, when $K_p$ tends to 0, the

avalanche moves but keeps its initial form. In terms of inertia, lower $K_p$ values indicate a lower momentum and lower

velocity.

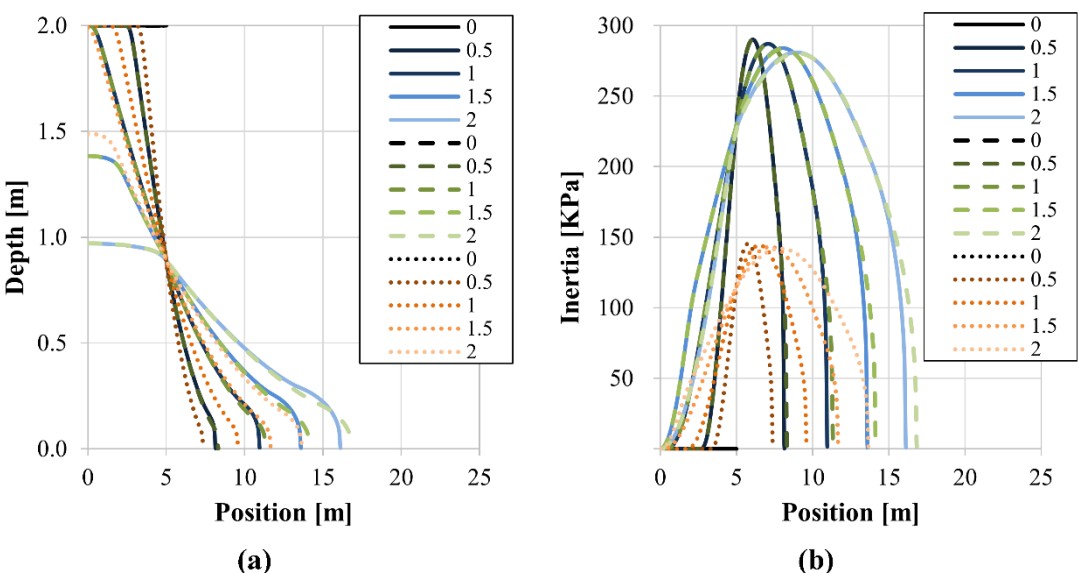

(a)                                                      (b)

**Figure 12. Effect of the $K_p$ factor on the flow behaviour of water (blue lines) and snow with $K_p = 1$ (green dashed lines) and $K_p = 0.5$ (brown dotted lines). (a) Free surface and (b) inertia evolution at the first 2 s, with intervals of 0.5 s in a dummy case study that represents a dam break.**




**Figure 13. Effect of the $K_p$ factor on the flow for experiment 9 of Bartelt et al. (2015). Observed and simulated results of the shear stress (a), flow depth (b) and velocity (c) by Bartelt et al. (2015) and Iber with $K_p = 1$ (1) and $K_p = 0.1$ (2).**





It is well known that, in nature, during the first time steps of most slab avalanches, snow moves like a block. However, in the simulations of avalanches this flow pattern usually is not properly reproduced. Most numerical tools reproduce a fast

fluidization in the initial steps, which can be quite different to what happens in nature. To solve this, many researchers consider an artificial initial condition: a snow mass with a constant velocity located at a short distance downstream of the release area (Ancey et al., 2004; Bartelt et al., 1999, 2015; Dent and Lang, 1980; Lang and Dent, 1980). Nevertheless, with a $K_p$ close to 0 at the initial steps, it is possible to represent the solid-like behaviour of the avalanche near the triggering area and thus there is no need to alter the initial conditions.

In order to highlight that, the flow behaviour of Experiment 9 presented by Bartelt et al. (2015) under different assumptions was analysed. The experiment consists in a 2.5 m wide straight chute with three different slopes. The results were analysed in terms of shear stress for $K_p$ values of 1 and 0.1. Figure 13a displays the evolution of the measured shear stress (black squared line) and that simulated by Bartelt et al. (2015) (dot-line) compared to the Voellmy–Salm total shear stress (red line) simulated with Iber. The Coulomb ($\mu$) and turbulent ($\xi$) contributions are also represented. Different tests were also

performed considering different $\xi$ values of 250, 500, and 1,000 m s$^{-2}$. In these cases, only the depth (Figure 13b) and the velocity (Figure 13c) evolution are plotted, which exhibit differences in the arrival time of the avalanche, as a consequence of the velocity reduction. The results of the shear stress better match the observations for $K_p = 0.1$ compared to $K_p = 1$ (Figure 13a). When $K_p = 0.1$, the snow arrives at the measuring point with a block-like shape, with higher depths and lower spreading (Figure 13b2) and sharper velocity profiles, better adjusted to the velocity trend of the observed data (Figure

13c2). Both numerical models considerably differ regarding the velocity field, but the velocity obtained with Iber follows a more similar trend to the observations, even with no initial condition impositions (simulation with Iber of the complete experiment). The general underestimation in terms of flow depth and shear stress can be attributed to uncertainties in the initial conditions: in the reference the starting volume is not stated clearly, but from its references the value of 13 m$^3$ was inferred; however, for a good fit in terms of shear stress, an initial volume of 17.6 m$^3$ was required.

This analysis indicates the importance of considering a nonhydrostatic pressure distribution on the mechanics of snow avalanches. However, more accurate observations and research are needed to better understand this complex phenomenon and provide recommendations for its modelling.

### 4.3. On Voellmy–Salm and cohesion models

It has already been seen that the values $\mu$, $\xi$, and $C$ can vary within a wide range. In most cases, these parameters are

considered to be constant in time and usually also uniform in space. Nevertheless, Bartelt et al. (2015) suggested that $\mu$ and $\xi$ may vary within the volume of the avalanche and over time as well. Variations of these coefficients, especially $\xi$, which has a wider range, could also be associated with the variations of snow characteristics, such as the density and internal moisture. However, it is not easy to define this link between the snow properties and the turbulent friction stresses.





For water flow, the friction stresses are usually calculated using the Manning formula and, thus, depend on the value of the
Manning coefficient ($n$). The Manning formula has also been used for the estimation of friction stresses for flows other than
that of water (Hungr, 1995; Ruiz-Villanueva et al., 2019). Ruiz-Villanueva et al. (2019) suggested a relationship between $\xi$
and $n$ as follows:

$$\xi = \frac{h^{1/3}}{n^2} \tag{9}$$

For example, as stated by Bartelt et al. (2017) for a tiny avalanche with a flow depth of 1 m and a return period of 30 years,
$\xi = 1{,}750$ m s$^{-2}$ is equivalent to a Manning coefficient of 0.024 s m$^{-1/3}$, which, according to Barnes (1987), corresponds to a
bottom surface formed by cobbles and gravel. In the same way, according to Arcement and Schneider (1989), a $n$ value of
0.1 s m$^{-1/3}$ that can be associated with a forested area, would correspond to a turbulent friction of 100 m s$^{-2}$, also for a snow
depth of 1 m. However, this last value is outside the range of values proposed by Bartelt et al. (2017), and the range
commonly found in the literature. From this analogy with the Manning coefficient, it is possible to relate the turbulent
friction coefficient ($\xi$) with a land-use cover. For example, when $\xi < 400$ m s$^{-2}$, which corresponds to "forested area"
according to Bartelt et al. (2017), $n$ would reach up to 0.05 s m$^{-1/3}$ for $h = 3$ m, but lower values of $h$ would reduce $n$.
Instead, if $\xi$ is between 400 and 1,000 m s$^{-2}$, as suggested by Rudolf-miklau et al. (2015), $n$ would range between 0.02 and
0.06 s m$^{-1/3}$. Furthermore, a $\xi$ value greater than 1,000 m s$^{-2}$, which is widely used for modelling snow avalanches (Christen
et al., 2010; Dreier et al., 2014; Fischer et al., 2009; Gruber and Bartelt, 2007; Schaub et al., 2016), would correspond to $n$
values lower than 0.04 s m$^{-1/3}$ that would decrease asymptotically to 0.02 s m$^{-1/3}$.

In hydraulics and hydrology the Manning coefficient is widely used and has been extensively studied and the association of a
roughness coefficient with land use or land cover is well defined. Thus, assessing the turbulent coefficient ($\xi$) from the
Manning coefficient ($n$) might also be interesting, because $\xi$ could be defined as a first approximation from on-earth or
remote sensing data of land use (e.g. CORINE land cover). Additionally, many authors have defined values for $n$ according
to land and flow characteristics (Bladé et al., 2019; Cea and Bladé, 2015; Ruiz-Villanueva et al., 2019; Sanz-Ramos et al.,
2018b, 2018a), especially for vegetated areas (Ebrahmimi et al., 2008; Green, 2005; O'Hare et al., 2010). Considering this
extensive knowledge, the relationship between $n$ and $\xi$ can be interesting for the estimation of this last parameter in the case
of full-depth avalanches, where the flow interacts directly with the terrain surface. For avalanches running over a snow layer,
there are a large number of experiments on flumes with artificial bottom roughness, both theoretical (Bouchet et al., 2003,
2004; Dent and Lang, 1982; Hutter et al., 1995; Lang and Dent, 1980) and with real snow (Bartelt et al., 2015; Jaedicke et
al., 2008; Lang and Dent, 1983; Platzer et al., 2007a; Rastello and Bouchet, 2007; Rognon et al., 2008), aiming to analyse
the snow–snow contact. This contact may vary during an experiment and between experiments, because of a restructuration
of the bottom, where the snow particles are flattened and cause the regularization of the bottom. More research is required on
the snow–snow contact friction surface in order to propose values of $\xi$ for this type of land use, considering the snow
characteristics (density, age, internal moisture, etc.).





On cohesion, the formulation proposed by Bartelt et al. (2015) [Eq. (6)] proved to be adequate to describe the effect of this

snow property in avalanche simulations, in terms of both stopping the avalanche and defining long tails. Tails are small snow

deposits composed by balls or clogs commonly stopped behind the avalanche front. However, using this formulation, the

influence of the cohesion term ($\tau_C$) in the results of 2D-SWE-based models is limited. $\tau_C$ is a function of cohesion ($C$),

Coulomb friction coefficient ($\mu$), snow density ($\rho$) and flow depth ($h$). Cohesion, a proportional factor, acquires values of

around 400 Pa up to 2,000 Pa (Bartelt et al., 2015; Dreier et al., 2014; Gaume et al., 2018; Platzer et al., 2007b). Gaume et

al. (2017) suggested a formula in which cohesion can be derived from the featuring energy of a slab-weak layer, following

the Mohr–Coulomb stability criteria. Furthermore, cohesion may vary before the avalanche release and after the avalanche is

stopped (Bartelt et al., 2015).

### 4.4. Implications for hazard assessments

The necessity of assessing natural hazards, such as snow avalanches, and of defining strategies to minimize the associated

risks has led to the development of ad hoc numerical models (Funk and Margreth, 1999; Gruber and Bartelt, 2007; Jamieson

et al., 2008; Keylock and Barbolini, 2011; Maggioni et al., 2019). In this line, several guides and technical documents on the

evaluation of the impact and the associated risks of snow avalanches have been published (CCA, 2016; Margreth, 2016;

Rudolf-Miklau et al., 2014), in which hazard assessments are based on the variables, mainly flow depth and velocity, that

result from numerical models. Thus, improving the techniques applied for the numerical modelling of avalanches will result

in more accurate hazard assessments.

In this line, the numerical treatment presented herein attempts to reproduce the dense-snow avalanche dynamics and

stopping without any nonphysically based assumption, thus avoiding the use of arbitrary parameters. Platzer et al. (2007b)

and more recently Bartelt et al. (2015) analysed the cohesion effects and proposed a formulation to considerate it in

numerical models. Gaume et al. (2019) investigated on modelling the snow release and flow in a coupled way, developing an

ad hoc numerical model for slab avalanches.

Moreover, in order to properly represent the snow avalanche dynamics, a good quality and high-resolution digital terrain

model (DTM) is mandatory. Maggioni et al. (2013) demonstrated the effect of the DTM on the results, especially when the

simulation was performed with the so called "summer topography" instead of "winter topography". Indeed, the summer

topography, despite being widely used because it is the only one available, applies only to full-depth avalanches. In addition,

numerical methods and schemes are also important to achieve good results.

Even with such improvements, further experimental analyses focused on determining the snow characteristics during the

whole phases of snow avalanches, including entrainment (which also modifies both the avalanche flow and the slope

surface), are still needed in order to provide physical descriptors of the snow dynamics and useful information to properly

calibrate and feed the numerical models.



## 5. Conclusions

Currently, a common technique for simulating non-Newtonian fluid dynamics, such as snow avalanches, is to solve the 2D mass and momentum conservation equations using together with a rheological model to adapt the friction terms to the fluid characteristics. The Voellmy–Salm model, which consists of Coulomb- and turbulent-type friction terms, is widely used for this purpose. Additional cohesion models have been more recently developed to represent the effect of cohesion during the triggering, release, propagation, and deposition steps of snow avalanches.

As friction is a relevant factor that is highly dependent on the avalanche dynamics, special care should be taken for its consideration in the numerical model, in accordance to the model's numerical scheme. In Iber, which is based on the finite volume method and the Roe numerical scheme, an upwind scheme has been used for Coulomb-type friction and cohesion, whereas for turbulent-type friction the numerical scheme uses a centred stencil. This ensures a correct balance between the convective terms of the equations and the friction terms, avoiding spurious oscillations of the free surface without the need for artificial viscosity or other stabilization techniques, even with complex geometries.

For physically-based model developers, the ideal model should be able to simulate the complete process of the physical phenomenon. There have been already some attempts to simulate the triggering/release/development/stopping of avalanches with a single tool, but the first steps of the avalanche motion, at which it has a block-like behaviour, are still challenging. Due to the equations used most commonly (mass and momentum), which are derived from the general equations of fluid motion (Navier–Stokes equations), the avalanche tends to have a fluid-like behaviour. Nevertheless, as it was proven herein, the hypothesis of nonhydrostatic anisotropic pressure can improve this model behaviour, provided that the $K_p$ parameter is well estimated.

The analysis of the friction–cohesion models revealed that the Voellmy–Salm model dominates the avalanche dynamics, whereas the cohesion model plays a relevant role in the definition of the avalanche tail. In agreement with other authors, it was found that the range of possible values for the governing parameters ($\mu$, $\xi$, and $C$) is very wide. As these parameters can be interpreted as calibration variables, it is possible to achieve good results when representing the avalanche dynamics with parameter values very far from those suggested in already existing guidelines. Understanding the role of the different parameters and terms of the equations in the avalanche evolution may favour model calibration, and it may contribute to making reasonable decisions when using numerical models for prognoses without field data.

Further research is still required to provide more efficient and reliable tools for snow avalanche modelling using the 2D approximation. Some important aspects that can still be improved are related to the numerical methods, the consideration of nonhydrostatic anisotropic pressure distribution, the snow entrainment, the treatment of topographic data (winter topography), and research leading to recommendations for parameter estimation, for example, from earth observations or satellite data (e.g., land cover databases). Finally, it is worth noting that several research groups have already presented or are developing new 3D modelling techniques (using both Lagrangian and Eulerian descriptions) with enhance model



capacities compared to 2D tools. However, 2D tools still have a promising future owing to their simplicity and computational performance.

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
