# Peer review of "Role of friction terms in two-dimensional modelling of dense snow avalanches"

_Natural Hazards and Earth System Sciences, 2019_

## Referee Comment (RC1) · Christophe Ancey (Referee) · 27 Apr 2020

The paper shows how the Iber numerical code (used in hydraulics) has been extended to cope with snow avalanches. It also presents three applications and discusses the part played by the various contributions to friction.

**Major comments**

This paper's strength lies in the extension of Iber to model snow avalanches. Iber is a freely available software based on efficient finite-volume techniques for solving the Saint-Venant equations, preprocessing and post-processing tools, and a user-friendly interface. Apart from commercial software such as RAMMS, existing tools are academic tools with no user interface, so Iber as a newcomer is welcome. The paper is

also interesting for two reasons:

- Developing numerical avalanche-dynamics models is a longstanding problem. To the best of my knowledge, most existing models are based on finite-volume techniques, following the idea proposed by Jean-Paul Vila in the 1980s (Vila, J.-P., Modélisation mathématique et simulation d'écoulements à surface libre. *La Houille Blanche,* 6/7, 485-489, 1984; Vila, J.P., Simplified Godunov schemes for 2*2 systems of conserved laws, *SIAM Journal of Numerical Analysis*, 23, 1173-1192, 1986. Vila, J.P., *Sur la théorie et l'approximation numérique des problèmes hyperboliques non-linéaires*, application aux équations de Saint-Venant et à la modélisation des avalanches denses, Ph.D. thesis thesis, Paris VI, 1986.). In the early 2000s, a benchmark comparison of numerical models showed how the numerical outcome was sensitive to the algorithm details (Barbolini, M., U. Gruber, C.J. Keylock, M. Naaim, and F. Savi, Application of statistical and hydraulic-continuum dense-snow avalanche models to five European sites, *Cold Regions Science and Technology*, 31, 133-149, 2000.). Today, 20 years later, if I compare my code based on clawpack (available from github) and Shaltop (developed by François Bouchut and Anne Mangeney), I got significant differences in the avalanche deposition zone in many cases. Developing new models and making them available should help us to improve the state of art, and see why (or when) some numerical approaches to the Saint-Venant equations are more efficient.

- Before the advent of commercial software (like Aval1d and Ramms), avalanche engineering was mostly the field of trained and experienced practitioners. The increasing availability of numerical tools has allowed a wider community of users (including untrained practitioners and governmental agencies) to access computational avalanche-dynamics models. Paradoxically, this has led to a significant decrease in the quality of expertise offered. Many people have been fooled by the apparent high resolution of numerical outcomes, confusing numerical resolution and prediction accuracy. Giving access to different avalanche-dynamics codes
should make people more aware of uncertainties affecting numerical simulations. As Bruno Salm stated in his last review paper, "The presented models are all—up to the present day—somehow uncertain. Therefore, only relative simple models with few parameters are significant. An increase of complexity of models does not necessarily mean an increase of accuracy or a better hazard mitigation strategy." (Salm, B., A short and personal history of snow avalanche dynamics, *Cold Regions Science and Technology*, 39, 83-92, 2004.)

That said, I think that the paper suffers from many shortcomings:

1. This paper's ultimate goal is unclear to me. The introduction does not frame any scientific issue. I understand that the authors want to study the effect of friction on the bulk dynamics, but I have hard time understanding what the problem is. Voellmy's model is an empirical one. It shows usefulness in many engineering applications, but there is no proof that snow behaves like a Voellmy frictional material (as shown in my 2004 JGR paper, Coulomb performs better in many cases). Although Adolf Voellmy did not present the issue like this, I presume that he was annoyed with Paul Mougin's model based on Coulomb friction (Mougin, P., *Les avalanches en Savoie*, 175-317 pp., Ministère de l'Agriculture, Direction Générale des Eaux et Forêts, Service des Grandes Forces Hydrauliques, Paris, 1922.) because an avalanche experiencing Coulomb friction cannot reach a steady state. The avalanche accelerates or decelerates. Hence, no possibility of providing analytical estimate of avalanche velocity. By adding a turbulent-like term, Voellmy got around this issue. To date, fitting the Voellmy coefficients or predicting avalanche behavior remains a difficult challenge. Adding new contributions to the Voellmy model would be justified if one can show that there is a clear advantage of using complex frictional models over simpler ones (Occam's razor). Comparison criteria (Brier skill score, Bayes factor, Akaike information, etc.) could help decide whether adding complexity is useful or not. When I see an empirical equation

like Eq. (2), I wonder how a model involving 5 dissipation sinks can perform better than simpler models like the Coulomb or Voellmy ones. I suggest revising the introductory material, framing general and specific issues, and specifying the scientific issue(s) addressed by the paper.

2. Section 2 needs refinement. The underpinning assumptions and governing equations should be clearly introduced. For instance, do the authors use a Cartesian frame? Curvilinear coordinates? The numerical algorithm used for solving the Saint-Venant equations should be written by keeping mind that the NHESS normal reader may not be familiar with Roe solvers. How the source term is taken into account or how the dry/wet limit is implemented needs to be fully specified.

3. Section 3 presents 3 case studies, and among them only the last one concerns a real-world avalanche. It would be interesting to include further comparison with well-documented avalanches, e.g. those monitored at La Sionne, Col du Lautaret, or Ryggfonn. Using high-resolution data (including front position over time, velocities, depth, etc.) would be useful to test Iber. A recent example of how field data can be used to deduced friction parameters is given by Heredia, M.B., N. Eckert, C. Prieur, and E. Thibert, Bayesian calibration of an avalanche model from autocorrelated measurements along the flow: application to velocities extracted from photogrammetric images, *Journal of Glaciology*, 1-13, 2020.

4. Section 4 contains overly general considerations on avalanche modelling. By focusing on a well-defined issue, applying Iber to several field cases, and discussing how prediction is improved by increasing the number of frictional parameters and how each frictional model performs relative to others would help beef up the discussion and dissipate the impression of rambling considerations.

I took a look at iberaula. I found the mention to Iber avalanche, but there is no information about the status of this code. Will it be available like Iber? Or reserved for collaborators, buyers, etc.?

Further work is required before the paper can be accepted for publication.

Christophe Ancey

**Specific comments**

1. L9: You probably confuse "Voellmy friction" and "Voellmy-Salm(-Gubler)" model. The latter is a computational method for estimating velocities and runout distances (the avalanche is assumed to behave like a sliding block experiencing Voellmy friction. The avalanche path is split into different parts, and on each part, the momentum balance equation is solved to provide the steady-state velocity.) See Salm, B., A. Burkard, and H. Gubler, *Berechnung von Fliesslawinen, eine Anleitung für Praktiker mit Beispielen*, Eidgenössisches Institut für Schnee- und Lawinenforschung (Davos), 1990. (Hansueli Gubler translated it into English or provided an English summary, if needed).

2. L28: I do not think that the Voellmy model is a "popular model" in the modelling of granular flows. It has mainly been used to model snow avalanches, and to a lesser extent debris flows.

3. L42: what do you mean with the effects of friction being ignored? Can you be more specific when you state that the parameters are nonphysical.

4. L49 a number of words (e.g. retention, detention, accretion, premise) throughout the paper seem to be used out of context.

5. Eq. (1) why do you use the delta symbol instead the partial differential operator. F is the flux function, not a tensor. And in Eq. (3) you do not show F, but its gradient.

6. L85: including snow entrainment into the governing equations involves modifying not only the mass balance equation, but also the momentum equation.

See for instance Iverson Ouyang (Entrainment of bed material by Earth-surface mass flows: review and reformulation of depth-integrated theory, Reviews of Geophysics, 53, 27-58, 2015) for a correct treatment of this problem. Many avalanche-dynamics models involving snow entrainment and deposition are inconsistent from the continuum mechanics viewpoint. The problem is complex (see Issler, D., Dynamically consistent entrainment laws for depth-averaged avalanche models, *Journal of Fluid Mechanics*, 759, 701-738, 2014; Ancey, C., and B.M. Bates, Stokes' third problem for Herschel-Bulkley fluids, *Journal of Non-Newtonian Fluid Mechanics*, 243, 27-37, 2017. Lusso, C., F. Bouchut, A. Ern, and A. Mangeney, A free interface model for static/flowing dynamics in thin-layer flows of granular materials with yield: simple shear simulations and comparison with experiments, *Applied Sciences*, 7 (4), 386, 2017.

7. Section 2.2: this section should describe the numerical methods more clearly. As the model uses the same numerical framework as Iber, it should focus on the papers by Bladé and Cea for the homogeneous equation, and describe more clearly how the source term is taken into account to correct the solution to the homogenous equation.

8. L190 probably better to place the information on the numerical parameters elsewhere

9. L209: Platzer measured the friction forces in a chute. There is no clear evidence that on a larger scale, the friction coefficient holds the same value (in the same way, in a granular packing, there is a weak link between particle friction and bulk friction).

10. L269 what do you mean with "a 2D model in the vertical"

11. L366 if the wet-dry limit is important, why do you mention it just here?

12. L420: the largest difference between simulated and real-world avalanches is that in the real world, an avalanche release is not like a dam break, in which a wall is suddenly removed. Initial rigidity or cohesion is probably a second-order problem, which does not influence the bulk dynamics significantly at later times.

13. L522: throughout the paper you have used 'physical' and 'non-physical', but these terms can be understood differently. You should be more specific.

---

## Author Comment (AC1) · 20 May 2020

Author's response to referee Christophe Ancey comments

COMMENT:

The paper shows how the Iber numerical code (used in hydraulics) has been extended to cope with snow avalanches. It also presents three applications and discusses the part played by the various contributions to friction.

Major comments:

This paper's strength lies in the extension of Iber to model snow avalanches. Iber is a freely available software based on efficient finite-volume techniques for solving the

Saint-Venant equations, preprocessing and post-processing tools, and a user-friendly interface. Apart from commercial software such as RAMMS, existing tools are academic tools with no user interface, so Iber as a newcomer is welcome.

ANSWER:

The authors want to firstly thank Mr. Ancey the time and dedication for reviewing the manuscript, besides his interest for Iber and its extension for simulation of dense-snow avalanches. Answers to each general comment are detailed below:

COMMENT:

The paper is also interesting for two reasons: Developing numerical avalanche-dynamics models is a longstanding problem. To the best of my knowledge, most existing models are based on finite-volume techniques, following the idea proposed by Jean-Paul Vila in the 1980s (Vila, J.-P., Modélisation mathématique et simulation d'écoulements à surface libre. La Houille Blanche, 6/7, 485-489, 1984; Vila, J.P., Simplified Godunov schemes for 2*2 systems of conserved laws, SIAM Journal of Numerical Analysis, 23, 1173- 1192, 1986. Vila, J.P., Sur la théorie et l'approximation numérique des problèmes hyperboliques non-linéaires, application aux équations de Saint-Venant et à la modélisation des avalanches denses, Ph.D. thesis thesis, Paris VI, 1986.). In the early 2000s, a benchmark comparison of numerical models showed how the numerical outcome was sensitive to the algorithm details (Barbolini, M., U. Gruber, C.J. Keylock, M. Naaim, and F. Savi, Application of statistical and hydraulic-continuum dense-snow avalanche models to five European sites, Cold Regions Science and Technology, 31, 133-149, 2000.). Today, 20 years later, if I compare my code based on clawpack (available from github) and Shaltop (developed by François Bouchut and Anne Mangeney), I got significant differences in the avalanche deposition zone in many cases. Developing new models and making them available should help us to improve the state of art, and see why (or when) some numerical approaches to the Saint-Venant equations are more efficient.

ANSWER:

The authors are in agreement with this general comment, and thank the provided bibliography. The aim of this paper is not to compare different numerical models or schemes, but the effects of the friction terms on the results when using one model. As the referee has already explained, there will be differences between the solutions. Herein two laboratory experiments are analysed, but only for Case 1 (Hutter, Exp. 117) the numerical experiments of Bartelt are shown, in order to emphasize the discrepancies regarding the role of the friction terms, through a brief comparison at the end of the section between the results obtained by the different models.

Before the advent of commercial software (like Aval1d and Ramms), avalanche engineering was mostly the field of trained and experienced practitioners. The increasing availability of numerical tools has allowed a wider community of users (including untrained practitioners and governmental agencies) to access computational avalanche-dynamics models. Paradoxically, this has led to a significant decrease in the quality of expertise offered. Many people have been fooled by the apparent high resolution of numerical outcomes, confusing numerical resolution and prediction accuracy. Giving access to different avalanche-dynamics codes should make people more aware of uncertainties affecting numerical simulations. As Bruno Salm stated in his last review paper, "The presented models are all—up to the present day—somehow uncertain. Therefore, only relative simple models with few parameters are significant. An increase of complexity of models does not necessarily mean an increase of accuracy or a better hazard mitigation strategy." (Salm, B., A short and personal history of snow avalanche dynamics, Cold Regions Science and Technology, 39, 83-92, 2004.).

ANSWER:

The authors are also in agreement with this comment. Numerical tools are a simplification of the reality, and the provided results, which also depend on the expertise of the technicians or modellers, should be taken carefully. This paper aims to show how

quite big changes in the friction terms allow to achieve very similar solutions. Section 3 shows that, which is an issue directly related to the friction terms expression, and not to the numerical code, as shown in Section 3.1. In particular this is shown with the Case study 2 (Section 3.3), in which different combinations of the friction parameters provide a good fit to the experimental results for the two experiments analysed (see Figure 8). The model used herein follows the same strategy for simulating dense-snow avalanches as other numerical codes: the use of Voellmy-fluid approximation for assessing the friction terms. There exist several codes based on the solution of the same equations, the 2D-SWEs. Thus, in all of them specific parameters and numerical strategies must be used, and is used, to avoid a water-like behaviour of the snow (see Sections 2.2 and 4.2). We are especially in agreement with Mr. Salm's comment, and the results presented herein show precisely the behaviour of a relative simple model with few parameters. From our point of view, there are two options for the numerical modelling of dense-snow avalanches: to use "simple" 2D-SWE based models; or to use much more complex Computational Fluid Dynamic (CDF) models. There is still a lack of expertise for the last ones, surely they are the future, but meanwhile it is worth exploring the capabilities and behaviour of the first. As we said before, we agree in the importance of expert criteria, and we added the following sentence in the Conclusions section: L536: . . . without field data. On the other hand, quality expert criterion is fundamental in the evaluation of the simulation outcomes.

COMMENT:

That said, I think that the paper suffers from many shortcomings: This paper's ultimate goal is unclear to me. The introduction does not frame any scientific issue. I understand that the authors want to study the effect of friction on the bulk dynamics, but I have hard time understanding what the problem is. Voellmy's model is an empirical one. It shows usefulness in many engineering applications, but there is no proof that snow behaves like a Voellmy frictional material (as shown in my 2004 JGR paper, Coulomb performs better in many cases). Although Adolf Voellmy did not present the issue like this, I

presume that he was annoyed with Paul Mougin's model based on Coulomb friction (Mougin, P., Les avalanches en Savoie, 175-317 pp., Ministère de l'Agriculture, Direction Générale des Eaux et Forêts, Service des Grandes Forces Hydrauliques, Paris, 1922.) because an avalanche experiencing Coulomb friction cannot reach a steady state. The avalanche accelerates or decelerates. Hence, no possibility of providing analytical estimate of avalanche velocity. By adding a turbulent-like term, Voellmy got around this issue. To date, fitting the Voellmy coefficients or predicting avalanche behavior remains a difficult challenge. Adding new contributions to the Voellmy model would be justified if one can show that there is a clear advantage of using complex frictional models over simpler ones (Occam's razor). Comparison criteria (Brier skill score, Bayes factor, Akaike information, etc.) could help decide whether adding complexity is useful or not. When I see an empirical equation like Eq. (2), I wonder how a model involving 5 dissipation sinks can perform better than simpler models like the Coulomb or Voellmy ones. I suggest revising the introductory material, framing general and specific issues, and specifying the scientific issue(s) addressed by the paper.

ANSWER:

Probably the goals of the manuscript are not clear enough. The authors focus the work on analysing the effects of the Voellmy–model parameters and cohesion on the avalanche characteristics, and its effect on mass and momentum equations. After the referees comment, the objectives section of the document will be restructured, with the aim to making them clearer, highlighting the ultimate goal as follows: The main aim of this work is a detailed analysis of the effects of the friction terms of the Voellmy-model and a cohesion-model on the results of the numerical modelling of dense-snow avalanche dynamics. The analysis focuses mainly on the influence of the friction–cohesion model on the determination of the shear stresses, and their effects on mass and momentum. The relevance and influence of each term has been tested by comparing the numerical results with well-documented laboratory experiments and with a real case study. Simulations were performed using the numerical tool Iber (Bladé et

al. 2014), a two-dimensional (2D) hydraulic model that has been recently enhanced to simulate dense-snow avalanches (Torralba et al. 2017). An additional aim of this work is to present the specific numerical treatment of the friction–cohesion model, that was implemented to adapt it to the particularities of the numerical scheme used by Iber: the Roe scheme (Roe 1986), which consists on the combination of the Godunov method together and the Roe Approximate Riemann Solver (Sanz-Ramos et al. 2020). The discussions on these numerical implementations and the understanding the role of the friction terms, together with some other considerations as the usage of nonhydrostatic pressure or nonisotropic properties, indicate that there is still a strong need for research on the description and modelling of the whole avalanche process (triggering, release, propagation, stopping, etc.). A last objective has been to provide contrasted data (numerical results) showing the final effects on the simulation of different avalanches (real and not) of variations in the involved parameters and procedures. This might provide criteria to modellers and avalanche risk analysis practitioners to better adjust the involved parameters and models options in some cases, or to decide on the need, or on the needlessness, of further parameter refinement, more detailed calibrations, or search for additional validation data. We also agree with the comments on there being no need for much complex models, and certainly, equation (2) shows a high degree of complexity is possible. Nevertheless, the document focuses in the Voellmy's model (Equation (5)) and not in all the possibilities underlying eq. (2). Regarding the friction models, certainly there are a lot of them, each one valid for the purposes that have been developed. Voellmy-fluid model is an empirical one, like other models in other fields, as for example sediment transport equations (Van Rijn, Meyer-Peter&Müller, Recking, etc.), and probably there are better ones, but it is widely used. Certainly, this model cannot reproduce the complex behaviour of the dense-snow avalanche dynamics. For this reason in Iber other friction models have also been included (Ruiz-Villanueva et al. 2019), widening its range of application, but this document is only focused in the Voellmy's and it being used for the simulation of dense-snow avalanches dynamics, thus the authors think that providing some information to assess its precision and uncertainties, as the role of the friction terms, can be helpful. On the other hand, the fact that in the Voellmy's model the friction terms are treated separately (solid phase and turbulent) has probably promoted it being used also for numerical modelling of granular flows also for other types of fluids than snow Hungr and McDougall (2009); Nam et al. (2019); Sartoris and Bartelt (2000); among others. Other approximations, as developed by Hutter, are also interesting (Hutter and Kirchner 2003). In order to clarify, Equation (2) only aims to show the theoretical different components of the shear stress, which could lead to much complex treatments of the frictional terms. Depending on the fluid behaviour, some shear stress terms can be more useful than others. This discretization is not analysed herein, and is neither implemented fully into Iber, being only the turbulent, the Mohr-Coulomb and the cohesion terms used for simulating the dense-snow avalanche in the model (Voellmy plus cohesion models). In this aspect, the authors have the feeling that among avalanche modellers there is a tendency to think that the Voellmy's model is especially adequate in this field (above other models), and that the commonly used parameters (from manuals and recommendations) are sometimes based on a consensus, perhaps based on legal reasons. As an additional objective, this work can give some light in showing that the parameters are not so unmovable (similar results can be obtained with different combinations) but also help in their determination as guidelines on the effects of each them are provided. The cohesion model recently proposed by Bartelt et al. (2015), Equation (6), has been included in the analysis because it as an additional friction term that can help to dissipate more energy and stop the avalanche, being cohesion an intrinsic property of the releasable snow. However, the model can work without considering cohesion terms (see Case 1, Section 3.2).

COMMENT:

Section 2 needs refinement. The underpinning assumptions and governing equations should be clearly introduced. For instance, do the authors use a Cartesian frame? Curvilinear coordinates? The numerical algorithm used for solving the Saint-Venant

equations should be written by keeping mind that the NHESS normal reader may not be familiar with Roe solvers. How the source term is taken into account or how the dry/wet limit is implemented needs to be fully specified.

ANSWER:

Due to the aims of the manuscript, in the original manuscript the numerical aspects of Iber were only briefly described, in Section 2.2, but highlighting the main differences with other numerical tools. We can find a similar criteria in other publications of NHESS (Ferrari et al. 2020; Franz et al. 2020; Kang and Kim 2019). The authors only want to remark the most important changes in the numerical scheme (Line 112), showing the benefits of using it against other methods. Nevertheless, below we add some extra information and description of the model, that will be includes, as far as it is possible for length and reading flux reasons, to the final document. Iber is a hydraulic numerical modelling tool that solves the 2D-SWE using the Finite Volume Method (FVM) in Cartesian framework. It uses an upwind first order Godunov scheme, specifically the Roe scheme (Godunov Method with the Approximate Riemman Solver of Roe), for the convective flux. It also uses un upwind discretization for the geometric slope source terms (Vázquez-Cendón 1999) and a first order centred scheme for the other source terms. In the Roe scheme, the decomposition of the integral of the flow vectors is performed using the eigenvectors of the Jacobian matrix. The equilibrium between the flux vector and the bed slope contribution of the source term (through its decomposition as a linear combination of the eigenvectors) allows avoiding spurious oscillations of the free surface when the geometry is complex (Bladé et al. 2012a; b; Bladé and Gómez-Valentín 2006; Brufau et al. 2002; LeVeque 2002; Toro 2009). With the Voellmy's model, an upwind scheme has been used for the friction terms of the solid-phase and the cohesion, which conceptually can be assimilated as an opposite to the slope step, which decelerates the flow when moving, or counterbalances the gravity forces when stopped (see Figure 1). The integration of the terms of solid friction and cohesion as part of the bottom slope terms can be interpreted as a "friction slope (Sanz-Ramos et

al. 2020a). The model uses the algorithm for the wet-dry front presented in Cea et al. (2007). The $\varepsilon$wd parameter defines the fluid depth threshold below which a finite volume (cell) is considered to be dry. When the fluid depth in a cell is lower than the threshold ($h < \varepsilon$wd), the finite volume is considered to be dry. The numerical treatment of $\varepsilon$wd only affects the computation of the mass and momentum fluxes, being the free surface and the flow velocity equal to zero when the bed elevation is higher than the free surface elevation (Cea and Bladé, 2015). This algorithm can successfully be used in finite volume schemes and ensures zero mass error (Brufau et al. 2004) . More information on the numerical scheme details can be found in the referenced bibliography (Bladé et al. 2012a; b, 2014b; Bladé and Gómez-Valentín 2006; Brufau et al. 2002; Cea and Bladé 2015; LeVeque 2002; Sanz-Ramos et al. 2020; Tan 1992; Toro 2009; Vázquez-Cendón 1999).

COMMENT:

Section 3 presents 3 case studies, and among them only the last one concerns a real-world avalanche. It would be interesting to include further comparison with well-documented avalanches, e.g. those monitored at La Sionne, Col du Lautaret, or Ryggfonn. Using high-resolution data (including front position over time, velocities, depth, etc.) would be useful to test Iber. A recent example of how field data can be used to deduced friction parameters is given by Heredia, M.B., N. Eckert, C. Prieur, and E. Thibert, Bayesian calibration of an avalanche model from autocorrelated measurements along the flow: application to velocities extracted from photogrammetric images, Journal of Glaciology, 1-13, 2020.

ANSWER:

The authors want to thank the referee to provide some well-documented real cases to compare it in deep with the simulated results from Iber. However, a deeper comparison with real data is far from the aims of the manuscript, which is focussed in analysing the friction parameters. The authors will consider those real cases for future works.

COMMENT:

Section 4 contains overly general considerations on avalanche modelling. By focusing on a well-defined issue, applying Iber to several field cases, and discussing how prediction is improved by increasing the number of frictional parameters and how each frictional model performs relative to others would help beef up the discussion and dissipate the impression of rambling considerations.

ANSWER:

Section 4 mainly aims to discuss some aspects, not enough considered previously, about of the limitations of the model because some of its hypothesis, either in the equations themselves or in the friction model, and how some further developments can improve the model's behaviour like consideration of non-hydrostatic pressure; more complex friction models (adding cohesion); and its implication on hazard assessment. All these aspects are related with the referee comment described in the Major comments. We think that the theoretical development and test cases show and clarify these aspects. The application to several field cases could be interesting, and as the flow behaviour is improved in the test cases, it will also be improved to the first ones. Nevertheless, due to the geometric simplicity and the no need of calibration, the last can help in better resenting and showing the discussed aspects.

COMMENT:

I took a look at iberaula. I found the mention to Iber avalanche, but there is no information about the status of this code. Will it be available like Iber? Or reserved for collaborators, buyers, etc.?

ANSWER:

The version for simulating dense-snow avalanches is not available to the public because it is currently under development. In this sense and before to open to the experts the avalanche module, our contribution also aims to show that Iber can simulate

snow avalanches as well as other numerical models; as a first step, we focussed on analysing the friction parameters and how they behave in Iber. As said before, a deeper comparison with real data is far from the aims of the manuscript. The authors will consider those real cases for future works. Nevertheless, as already happens with other modules under development, the authors welcome collaboration with other researchers and institutions. Iber is a final modelling tool for anybody to use it, but it is also an instrument for several research groups to be able to develop or test their own codes or methods, on the basis of an already existing framework. Collaboration, which is always welcome, may imply sharing the existing code, or just using it. As has already happened with other modules of Iber, once its robustness is proven, the module be freely distributed.

Specific comments:

PREVIOUS NOTE: the authors maintain the comments numbering used by the referee to answer it, and, in brackets, the corresponding line number of the PDF original manuscript.

COMMENT:

L9: You probably confuse "Voellmy friction" and "Voellmy-Salm(-Gubler)" model. The latter is a computational method for estimating velocities and runout distances (the avalanche is assumed to behave like a sliding block experiencing Voellmy friction. The avalanche path is split into different parts, and on each part, the momentum balance equation is solved to provide the steady-state velocity.) See Salm, B., A. Burkard, and H. Gubler, Berechnung von Fliesslawinen, eine Anleitung für Praktiker mit Beispielen, Eidgenössisches Institut für Schnee- und Lawinenforschung (Davos), 1990. (Hansueli Gubler translated it into English or provided an English summary, if needed).

ANSWER:

We used the notation "Voellmy-Salm" wrongly following similar notations found in some

literature. We will corrected it accordingly in the final document, and thank the reviewer for such clarification.

COMMENT:

L28: I do not think that the Voellmy model is a "popular model" in the modelling of granular flows. It has mainly been used to model snow avalanches, and to a lesser extent debris flows.

ANSWER (L32):

Probably this model is not the most popular model for debris flows, nevertheless there are several numerical tools that use this approach, including case studies with a reasonable fitting: Hürlimann, M., Rickenmann, D., Medina, V., and Bateman, A. (2008). "Evaluation of approaches to calculate debris-flow parameters for hazard assessment." Engineering Geology, Elsevier B.V., 102(3–4), 152–163. Schraml, K., Thomschitz, B., Mcardell, B. W., Graf, C., and Kaitna, R. (2015). "Modeling debris-flow runout patterns on two alpine fans with different dynamic simulation models." Natural Hazards and Earth System Sciences, 15(7), 1483–1492. Medina, V., Hürlimann, M., and Bateman, A. (2008). "Application of FLATModel, a 2D finite volume code, to debris flows in the northeastern part of the Iberian Peninsula." Landslides, 5(1), 127–142. Scheidl, C., Rickenmann, D., and McArdell, B. W. (2013). "Runout Prediction of Debris Flows and Similar Mass Movements." Landslide Science and Practice, Springer Berlin Heidelberg, Berlin, Heidelberg, 221–229. Rickenmann, D., Laigle, D., McArdell, B. W., and Hübl, J. (2006). "Comparison of 2D debris-flow simulation models with field events." Computational Geosciences, 10(2), 241–264. Nam, D. H., Kim, M. Il, Kang, D. H., and Kim, B. S. (2019). "Debris flow damage assessment by considering debris flow direction and direction angle of structure in South Korea." Water (Switzerland), 11(2), 1–16. Hungr, O., and McDougall, S. (2009). "Two numerical models for landslide dynamic analysis." Computers and Geosciences, Elsevier, 35(5), 978–992.) The authors indicated some of them in the manuscript (Hussin et al. 2012; Pirulli and Sorbino 2008;

Schraml et al. 2015), but nevertheless the final manuscript will be corrected accordingly to the reviewer's comment and the word "popular" will be dropped.

COMMENT:

L42: what do you mean with the effects of friction being ignored? Can you be more specific when you state that the parameters are nonphysical.

ANSWER:

The values of the frictional parameters of the Voellmy model have been widely discussed in the literature, there are even guidelines that can help, as the referee indicates, for "untrained practitioners" to choose them. Some of these values have been have been considered to be adequate because they provided good enough results for the purposes of a particular case study (e.g. snow avalanches in Alps), and in many cases further analysis of them, calibration, or even questioning if a certain value could be possible, has been subsequently omitted. These parameters should be within the range of application of the empirical equation, but as denoted in Section 3.1, and particularly with the case study presented herein, there can a wide range of parameters that provide similar solutions. Which one or which combination are the best? Why we choose ones instead others? How is the effect of basing the model on an adaptation of the SWE? These are the reasons we wanted to express with the sentence "[...] the effects of the friction model on the individual terms of the equations are commonly ignored [...]", but probably a better and more clarifying expression can be found. This line will be rewritten accordingly in the final version after all revisions and discussions, in order to make the message clearer. We differentiate between physical and non-physical based parameters of the equations, being the first those that can be a measurable property of the material and the latter the rest of parameters. Thus, the Coulomb friction coefficient ($\mu$) and the cohesion (C) can be considered two physically-based parameters that depend on the snow properties (Bartelt et al. 2015). But, we consider the turbulent friction coefficient ($\xi$) to be a non-physically based parameter.

In this line, the work of Fischer et al. (2015), shows an interesting analysis on this respect, suggesting that the effect of the turbulent friction coefficient is negligible for high values, in agreement with the results presented herein. This meaning of "physical" and "non-physical" will also be clarified in the updated manuscript that will be prepared after all the reviewers' revisions and discussions.

COMMENT:

L49 a number of words (e.g. retention, detention, accretion, premise) throughout the paper seem to be used out of context.

ANSWER:

The above-mentioned words are used in the following parts of the manuscript: L49, the word retention is in "[. . .] Additionally, Bartelt et al. (2015) proposed the inclusion of an additional friction term related to snow cohesion, a real physical snow property, which has an effect of retention and can stop the avalanche irrespective of the maximum momentum reached during the avalanche propagation [. . .]", and means that the cohesion is a property that can maintaining it aggregate, providing an "extra" force for holding it against the motion. L 59, the word detention is in "[. . .] The discussions on these numerical implementations, together with some other considerations like the usage of nonhydrostatic pressure or nonisotropic properties, indicate that there is still a strong need for research on the description and modelling of the whole avalanche process (triggering, release, motion, detention, etc.). [. . .]", and means the process of the stop of the avalanche. We will change for the word "deposition". L233, the word accretion is in "[. . .] An accretion of ðÌÌJŔðÌŘű with ℎ can be observed, more accentuated for lower values of ℎ; and a linear diminution of the shear stress, regardless of the flow depth, while ðÌÌJĞ increases [. . .]", and means an increment. We will change for the word "increment". L394, the word premise is in "[. . .] For water, the 2D-SWE usually assume a hydrostatic and isotropic pressure distribution (Chaudhry, 2008). This means a linear variation in the vertical direction with the specific weight of the flow and the

same in all horizontal directions. However, for non-Newtonian flows and steep slopes, this premise cannot be realistic (Ruiz-Villanueva et al., 2019). [. . .]", and it refers to the assumption of a hydrostatic and isotropic pressure distribution. We will change for the word "assumption".

COMMENT:

Eq. (1) why do you use the delta symbol instead the partial differential operator. F is the flux function, not a tensor. And in Eq. (3) you do not show F, but its gradient.

ANSWER:

In order to clarify the equation's notation, this part will be re-written following the most commonly notation used for 2D-SWEs, which in compact conservation form with source terms are:

(see Fig1.png)

where U is the conserved variable vector, F and G are the x and y components of the flow vector, and H is the source term. Momentum equations contain the gradients of the pressure and inertia terms (through the flow vectors F and G), the bottom slope and friction terms (through the source term H):

(see Fig2.png)

where h is the flow depth, $v\_x$ and $v\_y$ are the two velocity components, g is the gravitational acceleration, $S\_{(o,x)}$ and $S\_{(o,y)}$ are the two bottom slope components, and $S\_{(rh,x)}$ and $S\_{(rh,y)}$ are the two components of the rheological model. For water flows, the $K\_p$ factor is equal to 1 (hydrostatic pressure), and E is the a variation rate of the fluid column at a specific point, for example, a source or a sink in an open channel (Bladé et al. 2019), the rainfall/infiltration in hydrological modelling (Cea and Bladé 2015b), or the snow entrainment for avalanches (Eglit and Demidov 2005).

COMMENT:

L85: including snow entrainment into the governing equations involves modifying not only the mass balance equation, but also the momentum equation. See for instance Iverson Ouyang (Entrainment of bed material by Earth-surface mass flows: review and reformulation of depth-integrated theory, Reviews of Geophysics, 53, 27-58, 2015) for a correct treatment of this problem. Many avalanche-dynamics models involving snow entrainment and deposition are inconsistent from the continuum mechanics viewpoint. The problem is complex (see Issler, D., Dynamically consistent entrainment laws for depth-averaged avalanche models, Journal of Fluid Mechanics, 759, 701-738, 2014; Ancey, C., and B.M. Bates, Stokes' third problem for Herschel-Bulkley fluids, Journal of Non-Newtonian Fluid Mechanics, 243, 27-37, 2017. Lusso, C., F. Bouchut, A. Ern, and A. Mangeney, A free interface model for static/flowing dynamics in thin-layer flows of granular materials with yield: simple shear simulations and comparison with experiments, Applied Sciences, 7 (4), 386, 2017.

ANSWER:

The authors know about this problem, but in this document, this aspect is only mentioned in the introduction and it has not been considered in the numerical simulations presented herein. Certainly, the entrainment of new snow, which has lower momentum, slows down the flux of the bulk. In the Conclusions section, we remark some aspects that still need improvements, as the treatment of the entrainment. We have already started working in including snow entrainment in our model through a Master Thesis (Jordi Castelló i Sant 2020), and we'd like to thank the reviewer for the provided references and the indications, a very useful information for future works.

COMMENT: Section 2.2: this section should describe the numerical methods more clearly. As the model uses the same numerical framework as Iber, it should focus on the papers by Bladé and Cea for the homogeneous equation, and describe more clearly how the source term is taken into account to correct the solution to the homogenous equation.

ANSWER:

A more detailed description of the numerical scheme used in Iber, including the treatment of the homogeneous equation is described in Answer 2 of the "Major comments". We will modify the final manuscript accordingly maintaining the aims and reading flux of the document. Adding a very detailed description of the numerical aspects could probably confuse the reader and make the document not clear. The provided references can be used for further information on those aspects.

COMMENT:

L190 probably better to place the information on the numerical parameters elsewhere.

ANSWER:

We considered to include the numerical parameters in Section "2 Material and Methods", particularly in Section 2.3, because it is here where there is a general description of the Case study, particular characteristics, geometrical description and, also, the numerical discretization used in all cases. The authors think that that warrants the reading flux.

COMMENT:

L209: Platzer measured the friction forces in a chute. There is no clear evidence that on a larger scale, the friction coefficient holds the same value (in the same way, in a granular packing, there is a weak link between particle friction and bulk friction).

ANSWER:

Certainly, Platzer measured this parameter in a chute with real snow, or at least coming from it. However, it should be expected the Coulomb friction stress ($\mu$) to have limits, because it is a property of the snow. Release of dense-snow avalanches is impaired for high values of $\mu$. Probably, as suggested by Bartelt et al. (2012), this parameter is a function of other parameters, but, deepening in this fact would include more complexity

to a very complex problem, and beyond the scopes of the work.

COMMENT:

L269 what do you mean with "a 2D model in the vertical".

ANSWER:

With this expression we colloquially use we meant a "2D depth-averaged model", in which the depth averaging is performed "in the vertical" in comparison with 2D models that are averaged "in the horizontal" or in the flow width, as for example CE-QUAL 2D used in reservoir. In the final manuscript we will use the standard, and appropriate, expression "2D depth-averaged model".

COMMENT:

L366 if the wet-dry limit is important, why do you mention it just here?

ANSWER (L367):

The authors want to remark relevance of the wet-dry threshold when dealing with numerical modelling of snow avalanches without distorting the reading of the manuscript. The wet-dry parameter is introduced firstly where the Case Studies are presented (Section 2.3), and further explanation and discussion about this parameter is in the Section 4 (Discussion).

COMMENT:

L420: the largest difference between simulated and real-world avalanches is that in the real world, an avalanche release is not like a dam break, in which a wall is suddenly removed. Initial rigidity or cohesion is probably a second-order problem, which does not influence the bulk dynamics significantly at later times.

ANSWER:

The authors totally agree that avalanches, at initial time steps, do not behave as a dam

break. There are a great variety mechanisms that triger snow motion and approaches for their modelling, as precisely stated by the reviewer in some works (Ancey and Bain 2015). Real slab avalanches, for example, tend to have a block-like behaviour during the first time steps, disaggregating partially or completely after they have travelled some distance. This is precisely why the authors suggest a correction on the pressure terms through the parameter Kp, in order to make the results differ from those of a dam break. In a dam brake, pressure in the wet side of the front push water towards the dry side, with more pressure at the bottom of it, leading to the characteristic and well-known shape that water takes in that case. Limiting the pressure terms might improve this behaviour and make the results similar to those of dense snow. It is also true that the larger the internal cohesion of the snow layer, the more block-like behaviour the avalanche will have in its initial steps, but this cohesion can be lost immediately after the release and become less relevant for the bulk dynamics as commented by the reviewer. All this does not have a relevant influence on the global avalanche runoff, but using a 2D-SWE based model, the consideration of non-hydrostatic anisotropic pressure distribution can help in representing this block-like (or non-dam break like) behaviour. Anyway, as stated in the document, from our view point, more research is needed to couple avalanche triggering-release and motion, as for example the works of Gaume et al. (2019). Nevertheless, we will update the section of the article with some of these considerations.

COMMENT: L522: throughout the paper you have used 'physical' and 'non-physical', but these terms can be understood differently. You should be more specific.

ANSWER (L523):

The authors use this distinction in order to refer the parameters that are measurable, as for example the Coulomb friction ($\mu$), in comparison with non-measurable parameters, as for example the turbulent coefficient ($\xi$), or criteria, as the fact to stop the avalanche using a momentum criteria (Bartelt et al., 2017). To clarify that, in the final manuscript this will be clarified accordingly.

REFERENCES

Ancey, C., and Bain, V. (2015). "Dynamics of glide avalanches and snow gliding." Reviews of Geophysics, 53(3), 745–784. Bartelt, P., Bühler, Y., Buser, O., Christen, M., and Meier, L. (2012). "Modeling mass-dependent flow regime transitions to predict the stopping and depositional behavior of snow avalanches." Journal of Geophysical Research: Earth Surface, 117(1), 1–28. Bartelt, P., Valero, C. V., Feistl, T., Christen, M., Bühler, Y., and Buser, O. (2015). "Modelling cohesion in snow avalanche flow." Journal of Glaciology, 61(229), 837–850. Bladé, E., Cea, L., Corestein, G., Escolano, E., Puertas, J., Vázquez-Cendón, E., Dolz, J., and Coll, A. (2014). "Iber: herramienta de simulación numérica del flujo en ríos." Revista Internacional de Métodos Numéricos para Cálculo y Diseño en Ingeniería, CIMNE (Universitat Politècnica de Catalunya), 30(1), 1–10. Bladé, E., and Gómez-Valentín, M. (2006). Modelación del flujo en lámina libre sobre cauces naturales. Análisis integrado en una y dos dimensiones. Centro Internacional de Métodos Numéricos en Ingeniería. Monografía CIMNE no 97, Junio 2006. Bladé, E., Gómez-Valentín, M., Dolz, J., Aragón-Hernández, J. L., Corestein, G., and Sánchez-Juny, M. (2012a). "Integration of 1D and 2D finite volume schemes for computations of water flow in natural channels." Advances in Water Resources, 42, 17–29. Bladé, E., Gómez-Valentín, M., Sánchez-Juny, M., and Dolz, J. (2012b). "Source term treatment of SWEs using the surface gradient upwind method." Journal of Hydraulic Research, 50(4), 447–448. Bladé, E., Sanz-Ramos, M., Dolz, J., Expósito-Pérez, J. M., and Sánchez-Juny, M. (2019). "Modelling flood propagation in the service galleries of a nuclear power plant." Nuclear Engineering and Design, 352, 110180. Brufau, P., García-Navarro, P., and Vázquez-Cendón, M. E. (2004). "Zero mass error using unsteady wetting–drying conditions in shallow flows over dry irregular topography." International Journal for Numerical Methods in Fluids, John Wiley & Sons, Ltd., 45(10), 1047–1082. Brufau, P., Vázquez-Cendón, M. E., and García-Navarro, P. (2002). "A numerical model for the flooding and drying of irregular domains." International Journal for Numerical Methods in Fluids, 39(3), 247–275. Cea, L., and Bladé, E. (2015a). "A simple and efficient unstructured finite volume scheme for solving the shallow water

equations in overland flow applications." Water Resources Research, 51(7), 5464–5486. Cea, L., and Bladé, E. (2015b). "A simple and efficient unstructured finite volume scheme for solving the shallow water equations in overland flow applications." Water Resources Research, 51(7), 5464–5486. Cea, L., Puertas, J., and Vázquez-Cendón, M. E. M.-E. (2007). "Depth averaged modelling of turbulent shallow water flow with wet-dry fronts." Archives of Computational Methods in Engineering, Springer, 14(3), 303–341. Eglit, M. E., and Demidov, K. S. (2005). "Mathematical modeling of snow entrainment in avalanche motion." Cold Regions Science and Technology, 43(1–2), 10–23. Ferrari, A., Dazzi, S., Vacondio, R., and Mignosa, P. (2020). "Enhancing the resilience to flooding induced by levee breaches in lowland areas: a methodology based on numerical modelling." Natural Hazards and Earth System Sciences, 20(1), 59–72. Fischer, J. T., Kofler, A., Fellin, W., Granig, M., and Kleemayr, K. (2015). "Multivariate parameter optimization for computational snow avalanche simulation." Journal of Glaciology, 61(229), 875–888. Franz, M., Jaboyedoff, M., Mulligan, R. P., Podladchikov, Y., and Take, W. A. (2020). "An efficient two-layer landslide-tsunami numerical model: effects of momentum transfer validated with physical experiments of waves generated by granular landslides." Natural Hazards and Earth System Science. Gaume, J., van Herwijnen, A., Gast, T., Teran, J., and Jiang, C. (2019). "Investigating the release and flow of snow avalanches at the slope-scale using a unified model based on the material point method." Cold Regions Science and Technology, Elsevier, 168(June), 102847. Hungr, O., and McDougall, S. (2009). "Two numerical models for landslide dynamic analysis." Computers and Geosciences, Elsevier, 35(5), 978–992. Hussin, H. Y., Quan Luna, B., Van Westen, C. J., Christen, M., Malet, J. P., and Van Asch, T. W. J. (2012). "Parameterization of a numerical 2-D debris flow model with entrainment: A case study of the Faucon catchment, Southern French Alps." Natural Hazards and Earth System Science, 12(10), 3075–3090. Hutter, K., and Kirchner, N. (2003). Dynamic Response of Granular and Porous Materials under Large and Catastrophic Deformations - Lecture Notes in Applied and Computational Mechanics Volume 31. Jordi Castelló i Sant. (2020). "Enhancement and application of numerical methods

for snow avalanche modelling." Universitat Politècnica de Catalunya. Kang, S., and Kim, B. (2019). "Effects of coupled hydro-mechanical model considering two-phase fluid flow on potential for shallow landslides: a case study in Halmidang Mountain, Yongin, South Korea." Natural Hazards and Earth System Sciences. LeVeque, R. L. R. J. (2002). Finite Volume Methods for Hyperbolic Problems. Cambridge Texts in Applied Mathematic, Cambridge Univ. Press, Cambridge. Nam, D. H., Kim, M. II, Kang, D. H., and Kim, B. S. (2019). "Debris flow damage assessment by considering debris flow direction and direction angle of structure in South Korea." Water (Switzerland), 11(2), 1–16. Pirulli, M., and Sorbino, G. (2008). "Assessing potential debris flow runout: A comparison of two simulation models." Natural Hazards and Earth System Science, 8(4), 961–971. Roe, P. L. (1986). "A basis for the upwind differencing of the two-dimensional unsteady Euler equations." Numerical Methods for Fluid Dynamics II, K. W. Morton and M. J. Baines, eds., 59–80. Ruiz-Villanueva, V., Mazzorana, B., Bladé, E., Bürkli, L., Iribarren-Anacona, P., Mao, L., Nakamura, F., Ravazzolo, D., Rickenmann, D., Sanz-Ramos, M., Stoffel, M., and Wohl, E. (2019). "Characterization of wood-laden flows in rivers." Earth Surface Processes and Landforms, 44(9), 1694–1709. Sanz-Ramos, M., Bladé, E., Torralba, A., and Oller, P. (2020a). "Las ecuaciones de Saint Venant para la modelización de avalanchas de nieve densa." Ingeniería del agua, 24(1), 65–79. Sanz-Ramos, M., Bladé, E., Torralba, A., and Oller, P. (2020b). "Las ecuaciones de Saint Venant para la modelización de avalanchas de nieve densa." Ingeniería del agua, 24(1), 65–79. Sartoris, G., and Bartelt, P. (2000). "Upwinded finite difference schemes for dense snow avalanche modeling." International Journal for Numerical Methods in Fluids, 32(7), 799–821. Schraml, K., Thomschitz, B., Mcardell, B. W., Graf, C., and Kaitna, R. (2015). "Modeling debris-flow runout patterns on two alpine fans with different dynamic simulation models." Natural Hazards and Earth System Sciences, 15(7), 1483–1492. Tan, W. . (1992). Shallow Water Hydrodynamics. Elsevier Science. Toro, E. F. (2009). Riemann Solvers and Numerical Methods for Fluid Dynamics. Springer Berlin Heidelberg, Berlin, Heidelberg. Torralba, A., Bladé, E., and Oller, P. (2017). "Implementació d'un model bidimensional per a simulació d'allaus de

neu densa." V Jornades Tècniques de Neu i Allaus: Pyrenean Symposium on Snow and Avalanches, Ordino, Andorra. Vázquez-Cendón, M. E. M. E. (1999). "Improved Treatment of Source Terms in Upwind Schemes for the Shallow Water Equations in Channels with Irregular Geometry." Journal of Computational Physics, Elsevier, 148(2), 497–526.
* * *
$$\frac{\partial}{\partial t} U + \frac{\partial}{\partial x} F(U) + \frac{\partial}{\partial y} G(U) = H(U)\text{¤} \tag{1}$$

**Fig. 1.**

$$U = \begin{bmatrix} h \\ hv_x \\ hv_y \end{bmatrix}; \quad F = \begin{bmatrix} hv_x \\ hv_x^2 + K_p g \dfrac{h^2}{2} \\ hv_x v_y \end{bmatrix}; \quad G = \begin{bmatrix} hv_y \\ hv_x v_y \\ hv_x^2 + K_p g \dfrac{h^2}{2} \end{bmatrix}; \quad H = \begin{bmatrix} E \\ gh\left(S_{o,x} - S_{rh,x}\right) \\ gh\left(S_{o,y} - S_{rh,y}\right) \end{bmatrix} \quad (3)$$

**Fig. 2.**

---

## Short Comment (SC1) · 7 Jun 2020

Comment 1 Congratulations on this research and development of this very useful tool. It is of great interest for practitioners to spread the tools for granular flows analysis in a well-balanced way: realistic physically based and easy understand and use. (Line 541): I completely agree that 2D models are useful for avalanche hazard analysis and assessment.

Comment 2 It is very promising the implementation of these 3 friction terms in IBER, not only both original of Voellmy-Salm model. (Line 45): If I understood correctly, it could be said that the stopping criterion based on momentum (user-defined fraction of achieved maximum momentum as lower threshold) corresponds to a macroscopic point

of view, which stops whole calculation. In contrast, the additional friction term related to snow cohesion is a real physical snow property, which has an effect of retention and can stop the avalanche locally, where is needed due concavity or other issues along the avalanche path. Daring suggestion: the friction–cohesion model could be called Voellmy–Salm-Bartelt friction model. Even more, if we want to summarize the evolution in such avalanche models, it could be used also the name Voellmy-Salm-Gruber-Bartelt snow avalanche model.

Comment 3 It seems that cohesion term plays a main role where the slope changes rapidly. Is that right? In the paper, only a real avalanche case is tested. It will be very interesting to see further examples, and I suggest testing run-up problems, for instance with a protection dam. Keeping in mind the hydrological origin of IBER it will be of great interest the analysis of avalanche dynamics across a concrete dam placed on a channel in a gully part of the path still in steep terrain. These are typical solutions of the first half of XXth century in the Pyrenees. Under these conditions, the effect of the dam is the lamination of the flow pulse, not only the deposit of part of the mass. How is that reproduced by IBER? Does the cohesive-friction term play an additional effect of velocity reduction, that could be critical in this kind of configurations?

Comment 4 Another avalanche case that could be interesting and useful for testing the application of avalanche formulation in IBER is the catastrophic avalanche sequence in Sewell (Chile). The avalanches occurred in 1914, 1926, 1941 and 1944 offer different scenario: wet and dry snow conditions, run-up or deflection, etc. (Line 106): IBER uses a first-order Godunov-type upwind scheme for convective fluxes and the geometric slope source term, in particular the Roe scheme, and a centred scheme for the turbulent diffusion friction source term. Therefore, the scheme achieves balancing of the bottom slope source term with the flow tensor, thereby avoiding spurious oscillations of the free surface and retaining quiescent water even when working with complex irregular geometries. According to that, it will be interesting to explore if run-up problems need some variation in these schemes, or they are already properly solved.

Comment 5 (Line 106): "IBER solves the described 2D-SWE through a conservative finite volumes scheme and on unstructured meshes of triangles and quadrilaterals." This could be a stong point for IBER in avalanche analysis, allowing a better description of topography roughness, channels... in comparison to the raster based models.

Comment 6 (Line 101): "IBER was initially developed for hydrodynamic and sediment transport simulations... Iber has been recently enhanced to simulate snow avalanches and a specific numerical treatment of the friction–cohesion model was implemented to adapt it to the particularities of the numerical scheme used by Iber." As far as Iber is used for flooding risk analysis, it will be a natural way to make easier the implementation of debrisflood and debrisflow scenarios in such studies. Are you planning to apply Iber to debrisflow?

Comment 7 (Line 180): When you mention the Avalanche Database of Catalonia (BDAC) could be referred to the reference: https://www.researchgate.net/publication/318724068_THE_AVALANCHE_DATA_IN_THE_CATALAN_PYRENEES_20_YE https://www.researchgate.net/publication/318723626_AVALANCHE_MAPPING_IN_THE_CATALAN_PYRENEES_BALAN

Comment 8 Is Iber able to consider friction parameters varying along space/time? It could be a next step? For instance, in a wet avalanche, friction parameters increase at deposition zone... In case 3, could this fact improve the result reducing the lateral spreading in the east part of deposit?

Line 418: to reproduce slab avalanche it is also possible to introduce higher value for cohesion at initial steps?

Comment 9 Line 406: "1. The pivoting point of the free surface is the same for all simulations, maintaining the length and depth positions in approximately 5 m and 0.9 m, respectively." What is the sense of this pivoting point in Fig. 12? Is it related to the instability degree in the starting zone (balance of topographic slope and friction slope)? If Kp factor is influencing the inertia term at the begining, it could have a big influence on tiny avalanche simulation?

Comment 10 I've found 2 minor errors: Figure 13: the caption says kp=0,5 instead of 0,1 Line 430: xi=2000 is also considered and shown in the figure 13

Comment 11 Line 500: Gaume et al 2019 are implementing the Material Point Method, which is specially able to describe both the initial instability and movement generation, and also the large scale deformation along the path, considering 3D variability of variables. With 2D-SWE it is clearly not possible to deal with the initial part (describing the activation of movement) ... But it is not necessary for common hazard analysis, where instability is defined by the scenario and the interest is focussed at the bottom of the slope where facilities are placed.

---

## Author Comment (AC2) · 9 Jun 2020

Author's response to short comments of Marc Janeras:

COMMENT 1:

Congratulations on this research and development of this very useful tool. It is of great interest for practitioners to spread the tools for granular flows analysis in a well-balanced way: realistic physically based and easy understand and use.

(Line 541): I completely agree that 2D models are useful for avalanche hazard analysis and assessment.

ANSWER:

The authors want to firstly thank Mr. Janeras the time and dedication for reading the manuscript, besides his interest for Iber and its usefulness for practitioners. Regarding the specific comment: Line 541: As we indicated in https://doi.org/10.5194/nhess-2019-423-AC1, there are two possibilities to simulate avalanche dynamics: to use "simple" 2D-SWE based models; or to use much more complex Computational Fluid Dynamic (CDF) models. From our point of view, "simple" models can be good enough to analyse the hazard and risk due to dense-snow avalanche in most situations. This manuscript explores the capabilities and behaviour of 2D-SWE-based models for other applications out of the "common" fields, such us simulating the avalanche dynamics.

COMMENT 2:

It is very promising the implementation of these 3 friction terms in IBER, not only both original of Voellmy-Salm model.

(Line 45): If I understood correctly, it could be said that the stopping criterion based on momentum (user-defined fraction of achieved maximum momentum as lower threshold) corresponds to a macroscopic point of view, which stops whole calculation. In contrast, the additional friction term related to snow cohesion is a real physical snow property, which has an effect of retention and can stop the avalanche locally, where is needed due concavity or other issues along the avalanche path. Daring suggestion: the friction–cohesion model could be called Voellmy–Salm-Bartelt friction model. Even more, if we want to summarize the evolution in such avalanche models, it could be used also the name Voellmy-Salm-Gruber-Bartelt snow avalanche model.

ANSWER:

The frictions terms can be implemented in many ways (see Eq. 2), and in Iber it has been considered by Voellmy-Salm model and the new cohesion model presented by Bartlet, among others. The cohesion adds "more resistance" against to the fluid motion, but this friction model depends also on the solid-phase.

[Figure]

As we show in Section 3.2, Iber is able to stop the avalanche without considering the cohesion effect. The fully conservative numerical scheme used (see Section 2.2) ensures that when the tractive forces do not exceed the resisting ones the avalanche stops. This happens whether the cohesion is considered or not. Anyway, cohesion is a physical snow property that can be worth considering. Thus, it is not necessary to consider any additional stopping criterion, such as one based on the maximum momentum achieved in the bulk during the avalanche evolution.

As suggested, it could be interesting to grouping or re-named the frictional model that includes Voellmy-Salm friction model and the cohesion model of Bartelt. In the final version of the document the naming of the models will be re-considered, or at least a discussion on the different naming options will be included. Iber includes all the terms in the referred models.

COMMENT 3:

It seems that cohesion term plays a main role where the slope changes rapidly. Is that right? In the paper, only a real avalanche case is tested. It will be very interesting to see further examples, and I suggest testing run-up problems, for instance with a protection dam. Keeping in mind the hydrological origin of IBER it will be of great interest the analysis of avalanche dynamics across a concrete dam placed on a channel in a gully part of the path still in steep terrain. These are typical solutions of the first half of XXth century in the Pyrenees. Under these conditions, the effect of the dam is the lamination of the flow pulse, not only the deposit of part of the mass. How is that reproduced by IBER? Does the cohesive-friction term play an additional effect of velocity reduction, that could be critical in this kind of configurations?

ANSWER:

The analysis of the friction–cohesion models revealed that the Voellmy–Salm model dominates the avalanche dynamics, whereas the cohesion model plays a relevant role in the definition of the avalanche tail.

The authors want to thank the interest of simulating different casuistry with Iber, such as dam-like protections. Iber, as a hydraulic model, can simulate dam breaks (Sánchez-Romero et al. 2019; Sanz-Ramos et al. 2019) and, thus, the effectiveness of a protection dyke in terms of reduction of the "peak" discharge can be assessed. Surely the cohesion term will affect the inertia terms reducing the peak, but it cannot be stated that in general the cohesion term plays a critical effect on velocity reduction, more than the other friction terms. Deeper analysis and comparison with theoretical and real case studies is advisable. Some analysis of protection structure effects have been already carried out in Flumen Institute (Torralba-Conill 2017), and also some cases with avalanche run up (Castelló 2020), but these are not the aims of the manuscript, which is focussed in analysing the friction parameters. The authors will consider this casuistic for future works. We expect to continue working on that.

COMMENT 4:

Another avalanche case that could be interesting and useful for testing the application of avalanche formulation in IBER is the catastrophic avalanche sequence in Sewell (Chile). The avalanches occurred in 1914, 1926, 1941 and 1944 offer different scenario: wet and dry snow conditions, run-up or deflection, etc.

(Line 106): IBER uses a first-order Godunov-type upwind scheme for convective fluxes and the geometric slope source term, in particular the Roe scheme, and a centred scheme for the turbulent diffusion friction source term. Therefore, the scheme achieves balancing of the bottom slope source term with the flow tensor, thereby avoiding spurious oscillations of the free surface and retaining quiescent water even when working with complex irregular geometries. According to that, it will be interesting to explore if run-up problems need some variation in these schemes, or they are already properly solved.

ANSWER: The authors want to thank for the suggestion of a well-documented real case to compare it in deep with the simulated results from Iber. We will consider this

real case for future works.

Line 106: the numerical scheme used in Iber was developed focusing into avoid numerical instabilities, balancing the source term and the flow vectors (further information can be found in Sanz-Ramos et al. 2020). In the previous reference, some theoretical case studies were simulated and, as it also shown in this manuscript. The scheme can deal with run up problems with no numerical instabilities or scheme limitations. Obviously, the adjustment of the involved parameters is still critical, and the limitations on the model approximation (2D) are still there.

COMMENT 5:

(Line 106): "IBER solves the described 2D-SWE through a conservative finite volumes scheme and on unstructured meshes of triangles and quadrilaterals." This could be a strong point for IBER in avalanche analysis, allowing a better description of topography roughness, channels... in comparison to the raster based models.

ANSWER:

From its beginnings, Iber can discretize the domain by means of meshes formed by triangles, quadrilaterals or combination of both. But it is also able to generate raster-based meshes, and for sure combination of all of them thanks to the mesh generator integrated in GiD (Ribó et al. 1999) the pre and post-processing package used by Iber. There are also several possibilities not only to generate meshes, but also to import different king of meshes such as *.stl, raster GDAL, RTIN, etc. All this is indeed an advantage when representing complex geometries, or particularities of the users, widening the possibilities to carry out more realistic simulations.

COMMENT 6:

"IBER was initially developed for hydrodynamic and sediment transport simulations: : : Iber has been recently enhanced to simulate snow avalanches and a specific numerical treatment of the friction–cohesion model was implemented to adapt it to the

particularities of the numerical scheme used by Iber." As far as Iber is used for flooding risk analysis, it will be a natural way to make easier the implementation of debrisflood and debrisflow scenarios in such studies. Are you planning to apply Iber to debrisflow?

ANSWER:

The current version of Iber has some capabilities to simulate debris flows as it implements the Voellmy-Salm friction model, which has been used to carry out this kind of analysis. Iber additionally has implemented a simplified Bingham model and a Manning-like model (Ruiz-Villanueva et al. 2019). Although the authors focused the development of this non-Newtonian module in dense-snow avalanches, we also have in mind the possibility to widen the model to simulate this kind of fluids. Anyway, it has to be mentioned that the utility of a simulation software depends on its intrinsic capabilities (numerical methods, interface) but also in the expertise acquired by the user's community along time. The application of the tool to new fields has to be done with caution, progressively, and with a rigorous results analysis.

COMMENT 7:

(Line 180): When you mention the Avalanche Database of Catalonia (BDAC) could be referred to the reference:

https://www.researchgate.net/publication/318724068_THE_AVALANCHE_DATA_IN_THE_CATALAN_PYRENEES_20_YE

https://www.researchgate.net/publication/318723626_AVALANCHE_MAPPING_IN_THE_CATALAN_PYRENEES_BALAN

ANSWER:

We want to thank to provide these manuscripts, and we will include them in the references.

COMMENT 8:

Is Iber able to consider friction parameters varying along space/time? It could be a next step? For instance, in a wet avalanche, friction parameters increase at deposition

zone... In case 3, could this fact improve the result reducing the lateral spreading in the east part of deposit?

Line 418: to reproduce slab avalanche it is also possible to introduce higher value for cohesion at initial steps?

ANSWER:

Iber is able to vary the friction parameters along the space, but not in time. Implementing time-variations could be an improvement on one hand, but on the other it might increase the difficulty to carry out a simulation because of the need of calibration of the parameters describing this temporal variations.

The simulations of the case study of Coll de Pal (section 2.3.3 and 3.4) show, effectively, that the deposit is always shifted slightly to the east part. Although space-time variations of the friction-cohesion parameters could improve the solution, we attributed the differences to the assumptions on the release area (shape, extension, and depth) and the use of summer topography, which can retain snow in some areas, and would probably be smoother in winter topography.

Line 418: in the current version of Iber it is not possible to introduce an initial value of cohesion at initial steps.

COMMENT 9:

Line 406: "1. The pivoting point of the free surface is the same for all simulations, maintaining the length and depth positions in approximately 5 m and 0.9 m, respectively." What is the sense of this pivoting point in Fig. 12? Is it related to the instability degree in the starting zone (balance of topographic slope and friction slope)? If Kp factor is influencing the inertia term at the beginning, it could have a big influence on tiny avalanche simulation?

ANSWER:

[Figure]

Line 406: In this part of the manuscript we wanted to highlight the relevance of the pressure terms during the avalanche dynamics, but specially during the avalanche release, because the assumption of hydrostatic isotropic pressure distribution could not be realistic in this phase. Figure 12a shows a profile of the fluid elevation across its direction, and the pivoting point is a stable point that is "not affected" by the upstream and downstream conditions. With that point, and the shape of the free surface, we wanted to highlight that the snow-like fluid behaves in a similar way to water flows, even for Kp factors equal to 1 or 0.5, if only the turbulent friction contribution was considered. The Kp factor will affect more the fluid behaviour, the deeper the avalanches are (see Eq. 3), because this parameter directly affects the pressure terms, which depends on the square of the fluid depth (g•h2/2)

COMMENT 10:

I've found 2 minor errors: Figure 13: the caption says kp=0,5 instead of 0,1 Line 430: xi=2000 is also considered and shown in the figure 13

ANSWER:

Regarding the Figure's 13 caption, the Kp factor used for the simulations was 0.1, which is in accordance with the text.

Line 430: We will correct it accordingly.

COMMENT 11:

Line 500: Gaume et al 2019 are implementing the Material Point Method, which is especially able to describe both the initial instability and movement generation, and also the large scale deformation along the path, considering 3D variability of variables. With 2D-SWE it is clearly not possible to deal with the initial part (describing the activation of movement) ... But it is not necessary for common hazard analysis, where instability is defined by the scenario and the interest is focussed at the bottom of the slope where facilities are placed.

ANSWER:

The authors are in agreement with this comment. However, there are some possibilities to simulate the triggering and the release of the avalanche together with its propagation in the same model, as for example linking a plate or slab structural model (failure) with the hydraulic model (propagation). This is one of our research lines currently in development.

REFERENCES Castelló, J. (2020). "Enhancement and application of numerical methods for snow avalanche modelling." Master thesis. Universitat Politècnica de Catalunya. Barcelona, Spain.

Ribó, R., De Riera, M., and Escolano, E. (1999). GiD Reference Manual. Ed. CIMNE, Spain.

Ruiz-Villanueva, V., Mazzorana, B., Bladé, E., Bürkli, L., Iribarren-Anacona, P., Mao, L., Nakamura, F., Ravazzolo, D., Rickenmann, D., Sanz-Ramos, M., Stoffel, M., and Wohl, E. (2019). "Characterization of wood-laden flows in rivers." Earth Surface Processes and Landforms.

Sánchez-Romero, F. J., Pérez-Sánchez, M., Redón-Santafé, M., Torregrosa Soler, J. B., Ferrer Gisbert, C., Ferrán Gozálvez, J. J., Ferrer Gisbert, A., and López-Jiménez, P. A. (2019). "Estudio numérico para la elaboración de mapas de inundación considerando la hipótesis de rotura en balsas para riego." Ingeniería del agua, 23(1), 1–18.

Sanz-Ramos, M., Bladé, E., Torralba, A., and Oller, P. (2020). "Las ecuaciones de Saint Venant para la modelización de avalanchas de nieve densa." Ingeniería del agua, 24(1), 65–79.

Sanz-Ramos, M., Olivares Cerpa, G., and Bladé i Castellet, E. (2019). "Metodología para el análisis de rotura de presas con aterramiento mediante simulación con fondo móvil." Ribagua, 6(2), 138–147.

Torralba-Conill, A. (2017). "Implementation of a two‐dimensional model for simulating Snow Avalanches." Master Thesis. Universitat Politècnica de Catalunya. Barcelona, Spain.

---

## Referee Comment (RC2) · Anonymous Referee #2 · 14 Jun 2020

This paper aims to assess the role of friction parameters, notably cohesion, in snow avalanche dynamics simulations. Besides an analysis of the respective contributions of the different friction terms, numerical results are compared to physical data for three test cases spanning different scales, from lab experiments to a large-scale chute and to a real avalanche case. I would certainly agree with the authors that systematic studies to better constraint the use of avalanche models are strongly needed, in particular for hazard assessment applications. The models currently used in the community need stronger validations and benchmarking (see e.g., Issler et al., J. Glaciol., 2018), and the presented study offers interesting insights along this line.

Unfortunately, the paper does not do full justice to these valuable objectives, and thorough revisions would be needed to meet the required standards of scientific publica-

tions. Needed improvements concern several axes: (1) Better description of the conditions and parameters used in the simulations. Currently, one would certainly not be able to reproduce the obtained results with the information provided. (2) More in-depth physical discussion of the results, notably in regards to the relations between friction parameters and snow quality (wetness in particular). This issue, which is practically not covered in the paper, would probably constitute the most important takeaway of the paper from a snow science perspective. Without such discussion, the presented results remain essentially formal, and drawing general conclusions applicable beyond the selected test cases appears difficult. (3) Clarification of numerous unclear sentences and statements throughout the manuscript. (4) Improvement of several figures and captions.

I provide below a detailed list of main and technical comments, intended to help the authors in this revision task. Among these, comments 15 to 22 concerning the physical discussion of the results, are probably the most important.

Main comments

1/ Introduction. The literature review on numerical models for simulating avalanche propagation needs to be completed. Models based on Voellmy-Salm friction law or variants have been developed by numerous groups, e.g. among others (far from exhaustive list): Naaim, M., Durand, Y., Eckert, N., & Chambon, G. (2013). Dense avalanche friction coefficients: Influence of physical properties of snow. J. Glaciol., 59(216), 771-782. Naaim , M., Naaim-Bouvet, F., Faug, T., Bouchet, A. (2004). Dense snow avalanche modeling: flow, erosion, deposition and obstacle effects. Cold Reg. Sci. Technology, 39(2–3), 193-204. Sampl, P., Granig, M. (2009). Avalanche Simulation with SAMOS-AT. International Snow Science Workshop, Davos 2009, Proceedings. Pudasaini, S. P., and M. Krautblatter (2014). A two-phase mechanical model for rock-ice avalanches, J. Geophys. Res. Earth Surf., 119, 2272–229.

2/ Eq. (3). Strictly speaking, entrainment and deposition during the flow influence not

only the mass conservation equation, but also the momentum balance: see e.g. Naaim et al, Surv. Geophys., 2003.

3/ The cohesion model used in the paper (Eq. (6)) is pretty complex, and was obtained from fitting a limited number of data (Bartelt et al., J. Glaciol., 2015). Did the authors also consider simpler models, such as a constant cohesion (which would also be consistent with the data)? The current model produces an abrupt drop in the cohesion contribution to shear stress for low depth values (as seen in Fig. 4). Does this abrupt drop play an effective role in the simulation results? A dedicated sensitivity analysis of this issue would certainly be useful.

4/ The whole section 2.2 on numerical schemes, including Fig. 5, is pretty difficult to follow. I would suggest either providing more details and explanations in order to have a really self-contained presentation of these issues (maybe in a dedicated appendix), or either removing this section altogether and referring the readers to previous publications in which they can find the relevant information.

5/ P.6, l.159. The wet-dry limit is mentioned here for the first time, without being properly defined before. Since this numerical parameter appears to play an important role, as later discussed in section 4.1, it would need to be introduced earlier in the paper. The criteria used to select the value of this dry-wet limit in the different application cases should be explained.

6/ Sections 2.3.1 and 2.3.2. The initial conditions used in the simulations of case 1 (Hutter experiments) would need to be described more precisely (initial geometry of the granular mass). Similarly, the way the authors deal with the lateral walls of the channel (boundary conditions), for case 1 as well as for case 2, should be explained. From Figure 7, it seems that no lateral variations are observed in the simulation results (quasi-1D flow). Is this true? What is the added-value of using a 2D model in this case?

7/ Sections 2.3.2 and 2.3.3. The main characteristics of snow used in the experiments of case 2 should be recalled. In particular, the liquid water content is an important information for discussing the cohesion values later employed in the numerical simulations (see also comment 22). Same remark for case 3: can the authors provide information regarding the quality of snow involved in the simulated avalanche?

8/ P.9, l.210-211. It is doubtful that slush flows would be characterized by large values of the friction coefficient mu. In fact, the rheology slush flows is frequently assumed to obey viscoplastic models, i.e. without a friction contribution (e.g., Jaedicke et al., CRST, 2008). Hence, mentioning friction stresses up to 11,000 Pa for slush flows appears irrelevant.

9/ Section 3.1: Besides discussing the individual contributions of friction, "turbulence" and cohesion to the stress, it would be instructive to cross-compare these different contributions between one another. Figures showing which contribution dominates the overall behavior depending on flow height and velocity, typically, would certainly be interesting.

10/ Section 3.2: How exactly are the rear and front positions of the avalanches extracted from the simulations? Are the definitions used for these positions comparable with those employed in the study of Bartelt et al. (J. Glaciol., 1999) used as a reference?

11/P.14, l.296-300. The criteria to select the different simulations "that better approximate the observed results" should be clearly explained. Is the matching based on runout, flow height, flow velocity? In particular, one can expect the correlations found between the different friction parameters (Eqs (7) and (8)) to strongly depend on the number and choice of these criteria. What is then the robustness of these correlations? Don't they simply reflect an insufficient number of matching criteria?

12/Section 3.3. Still on the correlations between friction parameters: if the authors can demonstrate some general relevance to these correlations, the ranges of validity of relations (7) and (8) would need to be clearly mentioned. I do not understand what is meant by a "good adjustment even for values that were out of the already reported

range" (l. 303-304). Wouldn't it be possible to use similar functional forms (either linear or logarithmic) for adjusting the results of the two experiments? If not, are there any differences between the two experiments, in terms of physical characteristics, snow type, etc., that could explain these different results?

13/ Section 3.4. Please explain how the three scenarios analyzed in detail were selected.

14/ Section 3.4. The discussion of Figures 9 and 10 is not really clear. Are these two figures obtained with different models? Or just with different parameters? The authors also mention the "use of summer topography" as a possible explanation for the differences observed between the two figures. However, the actual topography used in the modeling is never indicated. And why using a different topography in the two cases? Finally, for the sake of comparison, it would be interesting to show velocity results also for the cases represented in Figure 9.

15/ Section 4.1. Besides the continuum assumption, one of the main assumption involved in 2D-SWE-based models is the shallow-flow assumption. The relevance of, and limitations implied by, this assumption would also need to be discussed in view of the different test cases considered in the paper.

16/ Section 4.2. Considering values of K_p different from unity allows one to consider anisotropic normal stresses in the material. The vertical stress does however remain "hydrostatic", ie linear with depth. I suggest modifying the title and discussions of this section accordingly.

17/ Figure 12. What is the friction law considered for water in this example? And what is the interest of only considering turbulent friction for the "snow" flows in this part? Since the comparison with water seems to add nothing to the discussion, I would actually suggest only showing results obtained for snow, with typical values of mu and xi and different values of K_p.

18/ Section 4.2. While the discussion concerning the capability of the model to represent block-like motion with low values of K_p is certainly interesting, the physical significance of such low K_p values would also need to be discussed in view of, e.g., classical active / passive theory in soils.

19/ P.22, l.435-437. The fact that Iber reproduces measured velocities better than Bartelt et al.'s model, is really not obvious in Figure 13. To me, both models actually appear to show considerable discrepancies with the measurements.

20/ P.22, l.438-439. The fitting performed on the volume of the avalanche should be clarified. If the flow volume considered in the two models is different, direct comparisons between the obtained results appear to loose much of their meaning.

21/ Section 4.3. The whole discussion about the possible relation between xi and Manning coefficient / roughness does not appear very relevant for avalanche applications, especially since a large part of the terrain roughness can be expected to be smoothed out in winter. The proposed analogy appears to be of little practical use, unless the authors can provide clear indications about the scale of the roughness to be considered.

22/ Section 4.3. Only a brief physical discussion of cohesion values is provided in this section, while this issue actually appears to me as the most interesting for avalanche applications. It is generally considered that dry snow can be represented as cohesionless, and that cohesion becomes important only for wet snow (e.g., Bartelt et al., J. Glaciol., 2015). However, in their simulations, the authors apparently applied cohesion values irrespective of snow quality. If the considered test cases only involve dry snow, one could question the relevance of including cohesion in the model. Can the authors provide arguments as to why cohesion would be needed also for dry snow? I strongly urge the authors to try and examine the role played by cohesion as a function of snow quality, and to add test cases involving wet snow if none is currently present.

Technical issues

- P.2, l. 42, "However, the effects of the friction model on the individual terms of the equations. . ." Unclear statement. Please consider rephrasing.

- P.3, l.70. Sentence is ambiguous, since dU/dt is also an inertia term.

- Different notations and decompositions are used throughout the paper for basal friction: tau_d, tau_t, tau_mc, etc. in eq. (2); S'_rh, S"_rh in eq. (4), tau_mu, tau_xi later on. This unnecessarily complicates the reading. Please homogenize these notations.

- P.3, l.83-84, and later. In fluid mechanics, pressure is generally defined an an isotropic component of the stresses. Hence, one should rather speak of non-isotropic normal stresses when K_p is different from unity.

- P.3, l.79-81. Related to the previous comment, the sentence starting by "Thus, if for water flow . . .", is not very clear.

- P.7, l.183. What is meant by "(stable condition)"?

- P.7, l.188. It would be useful to also indicate the total volume of the simulated avalanche.

- P.12, l.260. Typo: xi instead of mu.

- Figures 5 and 6. It would be clearer to use similar symbology in both figures, i.e. avoiding representing simulation results with discrete points in one figure and continuous curves in the other.

- Figure 5. The caption mentions different combinations of mu and xi, while only the value of xi is varied in the displayed results.

- Figure 6. The fact that very similar results are obtained with significantly different combinations of mu and xi appears surprising, and would certainly deserve to be commented in the text.

- P.12, l.268. "Bartelt et al., 1999, used a 2D model in the vertical." This formulation

is not very clear, as both Bartelt et al.'s and the present study use a depth-integrated model. The model of Bartelt et al. could be described as 1D (or 1.5D), whereas the present model is 2D (or 2.5D).

- P.12, l.270. Among the differences with the model used by Bartelt et al. (1999), one should also mention the use of anisotropic normal stresses with active/passive coefficients. In contrast, and although this is not clearly indicated in the paper, the authors only considered isotropic normal stresses for this application case. Can this difference explain the different behaviors observed in the results?

- Figure 7. Figures 7b and 7c are not very clear. A horizontal scale should be indicated. What do the different black lines represent?

- Figure 7. The exact definition of the "inertial forces" represented on Figure 7c should be given.

- P.15, l.308. What is meant by "a uniform estimation of the parameters throughout the model"?

- P.15, l.323-333. The fact that three scenarios are described in more detail should be explained prior to this paragraph. Otherwise, the transition with what precedes is hard to follow.

- P.17, l.342. Sentence starting with "Figure 10a shows the slope vectors" is unclear.

- Figure 12. To what do the different curves correspond? Different times? This should be explained.

- Figure 13 and related text. The values of mu and xi used in Figs. 13a should be indicated. Same for the value of mu in Figs. 13b and 13c. Also, the value of K_p indicated at the top of the right column appears to disagree with the caption and the text.

2019-423, 2020.

---

## Author Comment (AC3) · 14 Jul 2020

Author's response to the referee #2 comments:

GENERALITIES:

This paper aims to assess the role of friction parameters, notably cohesion, in snow avalanche dynamics simulations. Besides an analysis of the respective contributions of the different friction terms, numerical results are compared to physical data for three test cases spanning different scales, from lab experiments to a large-scale chute and to a real avalanche case. I would certainly agree with the authors that systematic studies to better constraint the use of avalanche models are strongly needed, in particular for hazard assessment applications. The models currently used in the community need

stronger validations and benchmarking (see e.g., Issler et al., J. Glaciol., 2018), and the presented study offers interesting insights along this line. Unfortunately, the paper does not do full justice to these valuable objectives, and thorough revisions would be needed to meet the required standards of scientific publications. Needed improvements concern several axes: (1) Better description of the conditions and parameters used in the simulations. Currently, one would certainly not be able to reproduce the obtained results with the information provided. (2) More in-depth physical discussion of the results, notably in regards to the relations between friction parameters and snow quality (wetness in particular). This issue, which is practically not covered in the paper, would probably constitute the most important takeaway of the paper from a snow science perspective. Without such discussion, the presented results remain essentially formal, and drawing general conclusions applicable beyond the selected test cases appears difficult. (3) Clarification of numerous unclear sentences and statements throughout the manuscript. (4) Improvement of several figures and captions. I provide below a detailed list of main and technical comments, intended to help the authors in this revision task. Among these, comments 15 to 22 concerning the physical discussion of the results, are probably the most important.

ANSWER:

The authors want to thank Referee #2 the time and dedication for reviewing the manuscript, besides sharing our vision regarding the necessity to carry out this kind of studies to improve the hazard assessment due to snow avalanches. The reviewer mentions four axes in which several improvements are required according to his expertise and opinion. The four axes involve mainly work of better description, more in depth discussion, clarification of sentences and figures and captions improvements. The authors are sure that these requirements can be fulfilled in the final version of the manuscript, and expect that the answers to the specific main comments and technical issues that follow will help to illustrate that.

MAIN COMMENT 1:

Introduction. The literature review on numerical models for simulating avalanche prop-agation needs to be completed. Models based on Voellmy-Salm friction law or variants have been developed by numerous groups, e.g. among others (far from exhaustive list): Naaim, M., Durand, Y., Eckert, N., & Chambon, G. (2013). Dense avalanche friction coefficients: Influence of physical properties of snow. J. Glaciol., 59(216), 771-782. Naaim, M., Naaim-Bouvet, F., Faug, T., Bouchet, A. (2004). Dense snow avalanche modeling: flow, erosion, deposition and obstacle effects. Cold Reg. Sci. Technology, 39(2–3), 193-204. Sampl, P., Granig, M. (2009). Avalanche Simulation with SAMOSAT. International Snow Science Workshop, Davos 2009, Proceedings. Pudasaini, S. P., and M. Krautblatter (2014). A two-phase mechanical model for rock-ice avalanches, J. Geophys. Res. Earth Surf., 119, 2272–229.

ANSWER: We want to thank the bibliography provided. We will thoroughly review the advances these documents represent and refer to them in the introduction, as well as follow them for other relevant literature, enlarging the introduction if necessary. A first view of the provided references shows that they are certainly of interest. In Naaim et al. (2004) a three parameter friction model is proposed, but in Naaim et al. (2013) the same authors perform a calibration using the Voellmy model (referring to it as "the standard two-parameter model") for an avalanche path in Chamonix valley. A detailed discussion of the Coulomb-Voellmy model can also be found in Pudasaini and Krautblatter (2014). Here, the authors remark that Coulomb-Voellmy type models are based on effectively single-phase dry granular flows, and as their purpose is to develop a two-phase model for rock-ice avalanches, they end using a model mainly based on friction angles. A difference approach is used by SAMOS Sampl and Granig (2009), which uses the SPH approach for the dense flow layer, with a bottom friction made of different terms which includes a yield stress, a Coulomb friction term affected by a fluidification factor, and a turbulent term. The 2D SPH approach assimilates each particle to a cylinder of height the avalanche depth, in comparison with the classical SPH approach where the particles are spheres (López et al., 2010; López Gómez et al., 2015) as in DAN3D (Schraml et al., 2015).

MAIN COMMENT 2:

Eq. (3). Strictly speaking, entrainment and deposition during the flow influence not only the mass conservation equation, but also the momentum balance: see e.g. Naaim et al, Surv. Geophys., 2003

ANSWER:

We totally agree. Certainly, the entrainment of new snow, with zero momentum, slows down the flux of the bulk. Anyway, in the document the entrainment is only mentioned in the introduction to present the context of the paper. About deposition, as the model is depth averaged, it takes place when in on element the total snow column has stopped, and as the governing equations are mass and momentum conservation, this happens when momentum becomes null. We have already started working in including snow entrainment in our model in the context of a Master Thesis (Castelló, 2020).

MAIN COMMENT 3:

The cohesion model used in the paper (Eq. (6)) is pretty complex, and was obtained from fitting a limited number of data (Bartelt et al., J. Glaciol., 2015). Did the authors also consider simpler models, such as a constant cohesion (which would also be consistent with the data)? The current model produces an abrupt drop in the cohesion contribution to shear stress for low depth values (as seen in Fig. 4). Does this abrupt drop play an effective role in the simulation results? A dedicated sensitivity analysis of this issue would certainly be useful.

ANSWER:

Certainly the cohesion model proposed by Bartelt et al. (2015) is quite complex, but we decided to include it in the model in order to assess its performance and its role in the frictions terms. We did not considered another cohesion model, as suggested by the Referee #2 (constant cohesion), but it is a quite interesting suggestion for future works in order to develop "simple" 2D-SWE models, in agreement with Salm (2004) and as

we mentioned in the Referee's #1 comments (https://doi.org/10.5194/nhess-2019-423-AC1).

The abrupt drop of this model is more abrupt for low values of the depth, but also for low values of cohesion (C). For values of C lower than 250 Pa, the cohesion contribution in the friction terms is only linearly dependent on the Coulomb friction coefficient (mu), so a model based on a linear approximation as a function of mu, could be also a good approximation. As we observed, the model proposed by Bartelt et al. (2015) plays an important role in the snow avalanche tail definition as it affects the stopping moment of the avalanche when its depth decreases. However, the general contribution of this friction model is quite limited in comparison with the other two terms (Coulomb friction and turbulent friction) due to the range of application of the parameters involved as we indicated in the last paragraph of section 4.3 of the manuscript.

MAIN COMMENT 4:

The whole section 2.2 on numerical schemes, including Fig. 5, is pretty difficult to follow. I would suggest either providing more details and explanations in order to have a really self-contained presentation of these issues (maybe in a dedicated appendix), or either removing this section altogether and referring the readers to previous publications in which they can find the relevant information.

ANSWER:

Following the suggestion of the Referee #2, and also to satisfy the interest of the Referee #1, we will include a full description of the numerical scheme as a supplementary material. Additionally, we will re-write this section, making it lighter, focusing on a conceptual description of the numerical scheme, but relegating the detailed mathematic expressions to the supplementary.

MAIN COMMENT 5:

P.6, l.159. The wet-dry limit is mentioned here for the first time, without being properly

defined before. Since this numerical parameter appears to play an important role, as later discussed in section 4.1, it would need to be introduced earlier in the paper. The criteria used to select the value of this dry-wet limit in the different application cases should be explained.

ANSWER:

We will modify the manuscript accordingly to include an extended definition of the wet-dry limit described in the Referee's #2 general comment 2. If when doing that the document exceeds the recommended length, we will include that as supplementary material. The criteria to select the wet-dry limit is mainly user-knowledge based, but it should consider the particle size that defines the flow. For this reason, we used a wet-dry limit equal to 0.003 m in the case study 1 (section 2.3.1), which is equal to the material size, and 0.01 m for the other cases because it is good enough for representing the extension and propagation of the flow. In any case, the model is able to simulate the flow propagation without numerical instabilities even using very low wet-dry limits (see for example Cea Gómez et al. 2020; Cea et al. 2007; Cea and Bladé 2015; Sanz-Ramos et al. 2018). The effect of the wet-dry limit for snow avalanches, is more on the results of the avalanche tail extension and thickness than on its propagation velocity, maximum depths and thus avalanche hazard. It is well known that for hydrological studies (with water) the effect of the wet-dry limit is more on the drying process than on the flood propagation (Bates and De Roo, 2000; Cea et al., 2007; Cea and Bladé, 2015; Grimaldi et al., 2018), which is consistent with the results for avalanches.

MAIN COMMENT 6:

Sections 2.3.1 and 2.3.2. The initial conditions used in the simulations of case 1 (Hutter experiments) would need to be described more precisely (initial geometry of the granular mass). Similarly, the way the authors deal with the lateral walls of the channel (boundary conditions), for case 1 as well as for case 2, should be explained. From Figure 7, it seems that no lateral variations are observed in the simulation results

(quasi-1D flow). Is this true? What is the added-value of using a 2D model in this case?

ANSWER:

For the experiment #117 (case 1) of Hutter et al. (1995), a constant elevation of snow was imposed on the release area warranting the same volume of the released material in both cases, physical and numerical experiment. We will modify the final version of the manuscript to clarify this aspect.

Regarding the wall conditions, in both cases we used "no-friction" conditions because the wall has a limited effect in comparison with the bottom roughness effect. However, Iber can compute the wall-boundary effects (see for example Bladé et al. 2014a; b; Cea and Bladé 2015).

Due to that, we decided to simplify the geometrical discretization of the Cases 1 and 2 in a 1D-mesh because the flow behavior is mainly 1D. In these cases, the lateral spread of the avalanches is negligible, so no important differences in the time-position relation of the avalanche are expected. For this reason we use a 1D mesh discretization, in which Iber solves the 2D-SWE but the Y component is zero.

MAIN COMMENT 7:

Sections 2.3.2 and 2.3.3. The main characteristics of snow used in the experiments of case 2 should be recalled. In particular, the liquid water content is an important information for discussing the cohesion values later employed in the numerical simulations (see also comment 22). Same remark for case 3: can the authors provide information regarding the quality of snow involved in the simulated avalanche?

ANSWER:

The aim of these tests is to calibrate the numerical model testing several combination of the friction terms in order to reproduce the study case, also assessing the friction terms and its effects in the flow propagation.

On one hand, for Case 2 (section 3.2.2), Dent and Lang (1980) only indicated that the snow "[...] had been sifted through a 6 mm wire mesh [...]". No additional information about the snow characteristics has been found for Case 2. On the other hand, the previous snow profiles and the post-avalanche study of the Case of Pal suggest that the snow was quite dry (no tests post-avalanche were performed). This avalanche occurred with relatively low temperatures (around -11oC) and a "quite cold" snow. The cohesion can be considered to be medium because the limit of the deposition area was clearly defined and some snow-aggregates were observed during the characterization of the avalanche (see attached Figure 1). The avalanche may be classified as a dense and dry with cohesion. However, the sudden change of the slope due to the road probably changed the avalanche properties from a dense to powder. The following video, which shows a controlled avalanche at the Coll de la Bonaigua in 2010 (Catalan Pyrenees range, northeast Spain), aims to illustrate this effect: https://www.youtube.com/watch?v=5yO-PSTKxCY. For this reason, as indicated, we tested several combinations of the friction parameters. As Referee #2 indicates, knowing the properties of the snow (liquid content, among others) can help to compare the results of the numerical simulations. For this reason, we will provide the characterization of the avalanche (Coll de Pal) as Supplementary material.

The authors want to notify a misprint when referencing the Case study 2. The correct reference of this experiment is Dent, J. D., and Lang, T. E. (1980). "Modeling of Snow flow." Journal of Glaciology, 26(94), 131–140. We will correct in the final version of the manuscript.

MAIN COMMENT 8:

P.9, l.210-211. It is doubtful that slush flows would be characterized by large values of the friction coefficient mu. In fact, the rheology slush flows is frequently assumed to obey viscoplastic models, i.e. without a friction contribution (e.g., Jaedicke et al., CRST, 2008). Hence, mentioning friction stresses up to 11,000 Pa for slush flows appears irrelevant

ANSWER:

Certainly, slush snow-avalanches should be treated with a different rheological model, such as a viscoplastic-based one. This fact is denoted in the results of the experiments of Platzer et al. (2007) when dealing with slush snow, in which the shear stresses are lower in comparison with wet or dry snow. We will re-write this paragraph accordingly as follows:

"[. . .] For high density values this limit can increase to 11,000 Pa. However, when dealing with high density snow, e.g. for slush snow that can reach a density of up to 750 kg/m3, the shear stress contribution is reduced (Jaedicke et al., 2008; Platzer et al., 2007a) because the fluid behavior can be similar to that of a viscoplastic fluid. [. . .]"

MAIN COMMENT 9:

Section 3.1: Besides discussing the individual contributions of friction, "turbulence" and cohesion to the stress, it would be instructive to cross-compare these different contributions between one another. Figures showing which contribution dominates the overall behavior depending on flow height and velocity, typically, would certainly be interesting.

ANSWER:

To carry out this cross-comparison and to represent this with a single graph involving the two or the three friction terms would be an excellent result, but it is a challenge, and even more if the figurer's objective is to clarify concepts. In terms of shear stress, on one hand, the Coulomb contribution and the cohesion contribution depend only on the flow depth and, on the other hand, the turbulent contribution depend only of the flow velocity. As we can observe in Figure 3, Coulomb and turbulent contribution are of the same order of magnitude for depths and velocities ranges from 0 to 2.5 m and from 0 to 40 m/s, respectively. Additionally, depth and velocity are correlated: high velocities imply low depths, and vice versa, for the same flow intensity. Indeed, for "fast" avalanches

the turbulent contribution dominates versus the others, while "slow" avalanches are the two other, Coulomb and cohesion, which dominates the shear stress contribution. Some authors (see Bartelt et al. 2015) suggest that mu and xi may vary within the volume of the avalanche and over time as well, increasing the complexity of analyzing these friction models. Considering that, we analyzed the different factors individually in order to simplify the analysis, and we think that the provided figures (Figure 3 and Figure 4) indicate, in an understandable way, the contribution of each parameter to the avalanche behavior, which is the main aim of the document.

MAIN COMMENT 10:

Section 3.2: How exactly are the rear and front positions of the avalanches extracted from the simulations? Are the definitions used for these positions comparable with those employed in the study of Bartelt et al. (J. Glaciol., 1999) used as a reference?

ANSWER:

In this section we numerically reproduced and compared the experiment described by Bartelt et al. (1999) in which, among other things, the time-position of the front and rear part of the avalanche are assessed. This experiment is based on the laboratory experiments of Hutter et al. (1995) in which, using high-speed photography, the evolution of the avalanche longitudinal profile was defined allowing "[. . .] to determine the position of the avalanche as a function of time [. . .]". In the numerical experiments we extract the position of the front and the rear part of the avalanche at the same time steps by observing the first and last element with a non zero snow depth.

MAIN COMMENT 11:

P.14, l.296-300. The criteria to select the different simulations "that better approximate the observed results" should be clearly explained. Is the matching based on runout, flow height, flow velocity? In particular, one can expect the correlations found between the different friction parameters (Eqs (7) and (8)) to strongly depend on the number

and choice of these criteria. What is then the robustness of these correlations? Don't they simply reflect an insufficient number of matching criteria?

ANSWER:

We carried out these simulations aiming to achieve good results in terms of the position of the leading-edge of the avalanche, mainly in the final position of the avalanche front at the deceleration area but also during the deceleration process (see Figure 7). Considering that, several combinations of the three friction parameters allowed to obtain a good fit between experimental and simulated results. The correlations (Eqs. (7) and (8)) extracted from the numerical results are an example of how using different parameters we can obtain similar results. This fact highlights the need to calibrate the numerical model, as well as the wide range of values of these parameters, besides combination of them, that allow to obtain very similar results. If field data had provided the values, or narrower ranges, for some parameters, the combinations to fit the final avalanche position would be less, but this case is a clear example of a practical case in which a calibration has to be carried out with the available parameters.

MAIN COMMENT 12:

Section 3.3. Still on the correlations between friction parameters: if the authors can demonstrate some general relevance to these correlations, the ranges of validity of relations (7) and (8) would need to be clearly mentioned. I do not understand what is meant by a "good adjustment even for values that were out of the already reported range" (l. 303-304). Wouldn't it be possible to use similar functional forms (either linear or logarithmic) for adjusting the results of the two experiments? If not, are there any differences between the two experiments, in terms of physical characteristics, snow type, etc., that could explain these different results?

ANSWER:

These correlations are extracted from two particular experiments, thus Eqs. (7) and (8)

are valid for reproducing them in the range shown in the Figure 8. When we indicate "[…] the already reported range […]" we refer to the range of values described at the beginning of the Section 3.3, which are: mu (0.1–0.3), xi (5,500–10,000 m/s2) and C (490–1,060 Pa). We carried out some other numerical experiments using these correlations, but for values out of the range previously indicated. The results shown also a good adjustment but with a lower R2. For that reason, at the end of this section we clarify that the application range valid for these experiments is "[…] limited to mu < 0.7 for Eq. (7) and for mu = xi = 0 for Eq. (8) […]". In any case, we will re-structure this section in the final manuscript to clarify how the equations were obtained and form which data.

Regarding the functional forms, we did not find any particular reason for experiment 1 and experiment 2 to be adjusted by a different functional form. No differences between the experiments are reported in terms of snow properties. The main difference was the terminal velocity achieved at the beginning of the deceleration area (12 m/s for Exp. and 18 m/s for Exp. 3). This fact probably modified the properties of the snow and, consequently, the relation between the parameters.

MAIN COMMENT 13:

Section 3.4. Please explain how the three scenarios analyzed in detail were selected.

ANSWER:

Once the 27 scenarios had been analyzed, we checked the fitting of observed data with the simulated in terms of run-out distance and amount of snow cumulated on the road. We finally selected 4 of 27 simulated scenarios (A1B1C2, A1B3C3, A2B2C1, A2B3C2) because these 4 had a run-out distance around 400 m and a snow depth on the road of around 2.4 m. The authors want to notify a misprint in the run-out distance, the correct one is 400 m (not 500 m such as indicated in the manuscript). We will modify the final version of the manuscript to correct this misprint and to clarify the selection of the analyzed scenarios.

MAIN COMMENT 14:

Section 3.4. The discussion of Figures 9 and 10 is not really clear. Are these two figures obtained with different models? Or just with different parameters? The authors also mention the "use of summer topography" as a possible explanation for the differences observed between the two figures. However, the actual topography used in the modeling is never indicated. And why using a different topography in the two cases? Finally, for the sake of comparison, it would be interesting to show velocity results also for the cases represented in Figure 9.

ANSWER:

The numerical model was the same for all scenarios (topography, initial conditions, etc.), we only modified the friction parameters. Figure 9 shows the final position of the avalanche for the 4 scenarios selected, which are discussed in the paragraph that starts in L323. Figure 10 shows the results of another parameter combination that follows the recommendations of Bartelt et al. (2017), besides the main directions of the topography (Figure 10a). For that, we use free topographical data available from the Catalan administration (Institut Cartogràfic i Geològic de Catalunya). The case study of Pal was not a "full-depth avalanche", so we think that it is important to highlight that the topography commonly provided, also used in this case, represents the "summer" topography, i.e. the elevation data without snow pack. In some cases, this fact could condition the flow propagation because the topography can be smoother or sharper depending on the previous meteorological and snowy conditions.

We will re-structure this section to clarify the referee's comments, also including the topographical source and the representation of the maximum velocity for the 4 analyzed scenarios (Figure 9).

MAIN COMMENT 15:

Section 4.1. Besides the continuum assumption, one of the main assumption involved

in 2D-SWE-based models is the shallow-flow assumption. The relevance of, and limitations implied by, this assumption would also need to be discussed in view of the different test cases considered in the paper.

ANSWER:

When referring to the two-dimensional Saint Venant Equations as the "Shallow Water Equations", the word "shallow " refers to the fact that the flow is indeed 2D. This, when developing the SWE equations from the 3D RANS equations, is considered when performing the depth averaging of variables. Thus the implications are that vertical components of the velocity are lost, and that a uniform velocity in the depth, in the longitudinal and transversal direction, are assumed. This is an intrinsic hypothesis of the equations that we used, and many other authors have been using, and certainly it is very interesting to analyze in which cases an avalanche flow can be reasonably simulated with this kind of equations. For example, powder avalanches that have an aerosol behavior cannot. Very shallow dense snow avalanches in most cases can (and that is why we refer to dense snow avalanches in the document). Anyway, the aim of the article is not to determine these ranges of validity, which should be part of the first chapters of avalanche dynamic modelling manuals, but to contribute to understand the role of the different terms involved in the simulations once the option to use SWE based models has been taken, whatever the reason is (perhaps as simple as the access to the tool).

MAIN COMMENT 16:

Section 4.2. Considering values of $K_p$ different from unity allows one to consider anisotropic normal stresses in the material. The vertical stress does however remain "hydrostatic", ie linear with depth. I suggest modifying the title and discussions of this section accordingly.

ANSWER:

Linear pressure distribution with the depth, and hydrostatic pressure distribution are not

synonyms. The general equation of hydrostatic pressure states that the gradient of the pressure (P) is proportional to the acting forces per unit of mass b(vector), being the proportionality constant the density of the fluid rho, that is rho X b(vector) = gradient(P). Thus, a value of Kp different than 1 means a non-hydrostatic pressure distribution, although it is linear. On the other hand, it is possible for the pressure distribution to be non-linear even in hydrostatic conditions (for compressible fluids). That is why we use the expressions "non-hydrostatic" and "anisotropic" along the document.

MAIN COMMENT 17:

Figure 12. What is the friction law considered for water in this example? And what is the interest of only considering turbulent friction for the "snow" flows in this part? Since the comparison with water seems to add nothing to the discussion, I would actually suggest only showing results obtained for snow, with typical values of mu and xi and different values of K_p.

ANSWER:

For the first example used in Section 4.2, we used a Manning-type friction law for water flow with Manning value equivalent to the turbulent friction coefficient obtained from Eq. (9). This part of the manuscript wants to highlight how using a 2D-SWE based model, even when modifying the pressure terms with a Kp factor, the flow has a water-like behavior. This only modification slightly improves the numerical representation at the first steps of the slab avalanches, but not the subsequent motion. We consider that this example contributes in highlighting the effect of the different terms of the equations on the avalanche movement, which is the main aim of the document.

MAIN COMMENT 18:

Section 4.2. While the discussion concerning the capability of the model to represent block-like motion with low values of K_p is certainly interesting, the physical significance of such low K_p values would also need to be discussed in view of, e.g., classical

active / passive theory in soils.

ANSWER:

The authors want to thank the referee suggestion. We agree with this comment and the necessity of comparisons with other theories. The physical significance of this low Kp can be inferred from the discussion in the document: at the first times steps, because of internal cohesion, snow moves as a rigid block. In a totally rigid block, as for example a solid, pressure is not transmitted through the mass, and thus Kp is zero. As snow is not totally rigid, Kp can be low but not zero, and indeed Kp should depend on the internal characteristics of the snow body (cohesion, layers structure, humidity, etc.). A comment in this line on this significance of the low values of Kp will be included in the final document. As the reviewer says, there is extensive literature on active/passive theory for soils.

MAIN COMMENT 19:

P.22, l.435-437. The fact that Iber reproduces measured velocities better than Bartelt et al.'s model, is really not obvious in Figure 13. To me, both models actually appear to show considerable discrepancies with the measurements.

ANSWER:

Certainly, neither of the two models could fully represent the behavior of snow in the channel experiment presented by Bartelt et al. (2015). However, the numerical results presented by Bartelt show that the bulk arrives 0.5 s faster at the measured point, while in Iber this gap is reduced to 0.2 s and 0 s for Kp = 1. For Kp = 0.1 respectively (xi = 2000 m/s2). Additionally, the simulated avalanche by Bartelt did not follow a "block-like" behavior because the flow depth and velocity decreases smoothly. In this way, the results presented herein show a more "block-like" behaves, especially when using low values of Kp, because the majority of the snow pass through the measuring point before the 2.5 s. In any case, with this case study the authors wanted to show the

benefits of using the Kp factor, and as we suggest at the end of the Section 4.2, more accurate observations and research are needed.

MAIN COMMENT 20:

P.22, l.438-439. The fitting performed on the volume of the avalanche should be clarified. If the flow volume considered in the two models is different, direct comparisons between the obtained results appear to loose much of their meaning.

ANSWER:

The initial volume is not stated in Bartelt et al. (2015), only "starting volumes" lower than 25 m3 are indicated in this reference. The original experimental campaigns were carried out by Platzer et al. (2007a, 2007b), who indicates that the initial volume was approximately 13 m3 for dry snow (Exp. 9 of Bartelt, used herein as an example for assessing different Kp values). So, we use this volume as initial condition. However, as we can observe in Figure 13a, the results of the shear stress were underestimated for both Kp evaluated, but specially for Kp = 1. For this reason, in order to achieve similar results in terms of shear stress we suggest that a greater initial volume can be required. We will modify the manuscript in order to clarify this aspect.

MAIN COMMENT 21:

Section 4.3. The whole discussion about the possible relation between xi and Manning coefficient / roughness does not appear very relevant for avalanche applications, especially since a large part of the terrain roughness can be expected to be smoothed out in winter. The proposed analogy appears to be of little practical use, unless the authors can provide clear indications about the scale of the roughness to be considered.

ANSWER:

The authors think that the analogy presented could be interesting from the point of view of calibration of the numerical model, in particular for full-depth dense-snow avalanches and for rare, large avalanches that penetrate into the forest. This correlation (Eq (9))

could help in the selection of the value of the turbulent coefficient (mu) as a first approximation, or especially when using spatially distributed mu values. The selection could be even automatized in a similar way to what is done with the Manning coefficient in hydraulics, associating it to the land uses.

MAIN COMMENT 22:

Section 4.3. Only a brief physical discussion of cohesion values is provided in this section, while this issue actually appears to me as the most interesting for avalanche applications. It is generally considered that dry snow can be represented as cohesionless, and that cohesion becomes important only for wet snow (e.g., Bartelt et al., J. Glaciol., 2015). However, in their simulations, the authors apparently applied cohesion values irrespective of snow quality. If the considered test cases only involve dry snow, one could question the relevance of including cohesion in the model. Can the authors provide arguments as to why cohesion would be needed also for dry snow? I strongly urge the authors to try and examine the role played by cohesion as a function of snow quality, and to add test cases involving wet snow if none is currently present.

ANSWER:

In the experiment presented herein, corresponding to Exp. 9 in Bartelt et al. (2015), a cohesion value of 396 Pa was used for all the simulations, the ones performed by the authors but also in the simulations presented by Bartelt. We did not discuss the necessity to consider or not the cohesion in this experiment, we just only applied it in order to compare the numerical results. In this section the analysis is focused in the relevance of Kp factor, not in the effect of the cohesion. However, the authors will consider the suggestion of the referee for future works.

TECHNICAL ISSUE 1:

P.2, l. 42, "However, the effects of the friction model on the individual terms of the equations. . ." Unclear statement. Please consider rephrasing.

ANSWER:

This sentence will be re-written as follows: "However, the effects of the friction model on the equations are commonly ignored."

TECHNICAL ISSUE 2:

P.3, l.70. Sentence is ambiguous, since dU/dt is also an inertia term.

ANSWER:

The description of the governing equation will be re-written as indicated in the general comment #5 of the Referee #1 (see https://doi.org/10.5194/nhess-2019-423-AC1).

TECHNICAL ISSUE 3:

Different notations and decompositions are used throughout the paper for basal friction: tau_d, tau_t, tau_mc, etc. in eq. (2); S'_rh, S"_rh in eq. (4), tau_mu, tau_xi later on. This unnecessarily complicates the reading. Please homogenize these notations.

ANSWER:

The authors want to remark that assuming the hypotheses of shear stress grouping (L74), the shear stress can be considered by different components. For example, the Voellmy friction model involves the turbulent term ($\tau$_t (xi)) and the Mohr-Coulomb term ($\tau$_t (mu)).

TECHNICAL ISSUE 4:

P.3, l.83-84, and later. In fluid mechanics, pressure is generally defined an isotropic component of the stresses. Hence, one should rather speak of non-isotropic normal stresses when K_p is different from unity.

ANSWER:

The manuscript will be corrected accordingly.

TECHNICAL ISSUE 5:

P.3, l.79-81. Related to the previous comment, the sentence starting by "Thus, if for water flow...", is not very clear.

ANSWER:

There is a misprint in L78. This sentence will be re-written as follows: "Thus, for water flow the shear terms due to friction are expressed by means of the friction slope [...]."

TECHNICAL ISSUE 6:

P.7, l.183. What is meant by "(stable condition)"?

ANSWER:

We refer to "stable conditions" for those scenarios where the snow pack does not generate an avalanche because the terrain slope is gentle (< 28o) or the terrain is not able to keep enough snow pack (> 45 o).

TECHNICAL ISSUE 7:

P.7, l.188. It would be useful to also indicate the total volume of the simulated avalanche.

ANSWER:

The total volume of the snow was not directly measured, but it is was estimated in approximately 2431 m3. The manuscript will be corrected accordingly.

TECHNICAL ISSUE 8:

P.12, l.260. Typo: xi instead of mu.

ANSWER:

The manuscript will be corrected accordingly.

TECHNICAL ISSUE 9:

Figures 5 and 6. It would be clearer to use similar symbology in both figures, i.e. avoiding representing simulation results with discrete points in one figure and continuous curves in the other.

ANSWER:

These figures will be modified accordingly.

TECHNICAL ISSUE 10:

Figure 5. The caption mentions different combinations of mu and xi, while only the value of xi is varied in the displayed results.

ANSWER:

The caption will be re-written as follows: Figure 5. Comparison between the measured positions (r: rear; f: front) by Hutter et al. (1995) and the computed results using mu = 0.49 different values of xi for Exp. 117. The lines represent the results of the computed simulation by Bartelt et al. (1999).

TECHNICAL ISSUE 11:

Figure 6. The fact that very similar results are obtained with significantly different combinations of mu and xi appears surprising, and would certainly deserve to be commented in the text.

ANSWER:

These results reinforce the hypothesis suggested by the authors, in which it is possible to achieve similar results in avalanche modelling with very different combinations of the parameters involved. We will add the following sentences: "[...] can be observed (Figure 6). This figure only represents the results of the combination best fit to the experiments, achieving similar results with different parameters combination. After t =

0.3 s, the velocity increases, resulting in a larger rear and front positions and further expansion of the avalanche. Small differences can be identified on the simulated rear part of the avalanche, whereas the runout of the front part decreased when mu and xi increased. [. . .]".

TECHNICAL ISSUE 12:

P.12, l.268. "Bartelt et al., 1999, used a 2D model in the vertical." This formulation is not very clear, as both Bartelt et al.'s and the present study use a depth-integrated model. The model of Bartelt et al. could be described as 1D (or 1.5D), whereas the present model is 2D (or 2.5D).

ANSWER:

With this expression we colloquially use, we meant a "2D depth-averaged model", in which the depth averaging is performed "in the vertical" in comparison with 2D models that are averaged "in the horizontal" or in the flow width, as for example CE-QUAL 2D used for reservoirs. In the final manuscript we will use the standard, and appropriate, expression "1D depth-averaged model".

TECHNICAL ISSUE 13:

P.12, l.270. Among the differences with the model used by Bartelt et al. (1999), one should also mention the use of anisotropic normal stresses with active/passive coefficients. In contrast, and although this is not clearly indicated in the paper, the authors only considered isotropic normal stresses for this application case. Can this difference explain the different behaviors observed in the results?

ANSWER:

The differences could also be due to the consideration of the active/passive coefficient that affects the pressure terms in the model proposed by Bartelt et al. (1999). In this case study we aimed to achieve good results comparing with the experimental data, without considering any additional hypothesis such as anisotropic pressure.

TECHNICAL ISSUE 14:

Figure 7. Figures 7b and 7c are not very clear. A horizontal scale should be indicated. What do the different black lines represent?

ANSWER:

The Figure 7b and 7c represent a plain view of the numerical model. The black lines represent a distance gap of 0.1 m. The figures will be modified accordingly.

TECHNICAL ISSUE 15:

Figure 7. The exact definition of the "inertial forces" represented on Figure 7c should be given.

ANSWER:

The results of inertia are derived directly from Eq. (3).

TECHNICAL ISSUE 16:

P.15, l.308. What is meant by "a uniform estimation of the parameters throughout the model"?

ANSWER:

We use "uniform" to indicate that all the parameters are were considered uniform spatially and temporally.

TECHNICAL ISSUE 17:

P.15, l.323-333. The fact that three scenarios are described in more detail should be explained prior to this paragraph. Otherwise, the transition with what precedes is hard to follow.

ANSWER:

The manuscript will be modified accordingly to properly introduce the scenarios that

are analyzed in detail.

TECHNICAL ISSUE 18:

P.17, l.342. Sentence starting with "Figure 10a shows the slope vectors" is unclear.

ANSWER:

This sentence will be re-written as follows: "Figure 10a shows the slope main directions of the terrain (vectors), which are in concordance with the RIT051 area (light blue polygon)".

TECHNICAL ISSUE 19:

Figure 12. To what do the different curves correspond? Different times? This should be explained.

ANSWER:

The different curves represented in Figure 12 correspond to the evolution of the free surface (12a) and the inertia (12b) of the described experiment (L403). The experiment duration is 2 s and the results are plotted each 0.5 s. The water fluid is represented by blue lines and the snow-like fluid with kp = 1 and kp = 0.5 is represented by green dashed lines and brown dotted lines, respectively. The text will be modify as follows:

[. . .] Figure 12a shows the free surface evolution during the 2 first seconds. For snow flow, only turbulent friction (xi = 1,600 m s-2), equivalent to the Manning coefficient for water flow, was implemented, with two different Kp values (1 and 0.5). [. . .]

And the caption as follows:

[. . .] Effect of the Kp factor on the flow behavior of water (blue lines) and snow with Kp = 1 (green dashed lines) and Kp = 0.5 (brown dotted lines). Evolution of the free surface (a) and the inertia (b) during the first 2 s, with intervals of 0.5 s in a dummy case study that represents a dam break. [. . .]

TECHNICAL ISSUE 20:

Figure 13 and related text. The values of mu and xi used in Figs. 13a should be indicated. Same for the value of mu in Figs. 13b and 13c. Also, the value of K_p indicated at the top of the right column appears to disagree with the caption and the text.

ANSWER:

The manuscript will be modified as follows: "[. . .] total shear stress (red line) simulated with Iber using the parameters suggested by Bartelt et al. (2015) (mu = 0.55 and xi = 2,000 ms-2). The Coulomb (mu) and turbulent (xi) contributions are also represented (dashed lines). Different tests were also performed considering different xi values of 250, 500, and 1,000 ms-2. [. . .]". Additionally, the misprint in the Kp for the figures 13b, 13d and 13e will be corrected accordingly.

REFERENCES

Bartelt, P., Salm, B. and Gruber, U.: Calculating dense-snow avalanche runout using a Voellmy-fluid model with active/passive longitudinal straining, J. Glaciol., 45(150), 242–254, doi:10.3189/s002214300000174x, 1999. Bartelt, P., Valero, C. V., Feistl, T., Christen, M., Bühler, Y. and Buser, O.: Modelling cohesion in snow avalanche flow, J. Glaciol., 61(229), 837–850, doi:10.3189/2015JoG14J126, 2015. Bartelt, P., Bühler, Y., Christen, M., Deubelbeiss, Y., Salz, M., Schneider, M. and Schumacher, L.: RAMMS: Avalanche User Manual, WSL Institute for Snow and Avalanche Research SLF. [online] Available from: https://ramms.slf.ch/ramms/downloads/RAMMS_AVAL_Manual.pdf, 2017. Bates, P. . and De Roo, A. P. .: A simple raster-based model for flood inundation simulation, J. Hydrol., 236(1–2), 54–77, doi:10.1016/S0022-1694(00)00278-X, 2000. Bladé, E., Cea, L., Corestein, G., Escolano, E., Puertas, J., Vázquez-Cendón, E., Dolz, J. and Coll, A.: Iber: herramienta de simulación numérica del flujo en ríos, Rev. Int. Métodos Numéricos para Cálculo y Diseño en Ing., 30(1), 1–10, doi:10.1016/j.rimni.2012.07.004, 2014a. Bladé, E., Cea, L. and Corestein,

G.: Modelización numérica de inundaciones fluviales, Ing. del Agua, 18(1), 68, doi:10.4995/ia.2014.3144, 2014b. Castelló, J.: Enhancement and application of numerical methods for snow avalanche modelling, Master thesis. Universitat Politècnica de Catalunya. Barcelona, Spain., 2020. Cea Gómez, L., Bladé i Castellet, E., Sanz-Ramos, M., Fraga Cadórniga, I., Sañudo Costoya, E., García-Feal, O., Gómez-Gesteira, M. and González-Cao, J.: Benchmarking of the Iber capabilities for 2D free surface flow modelling, Universidade da Coruña. Servizo de Publicacións., 2020. Cea, L. and Bladé, E.: A simple and efficient unstructured finite volume scheme for solving the shallow water equations in overland flow applications, Water Resour. Res., 51(7), 5464–5486, doi:10.1002/2014WR016547, 2015. Cea, L., Puertas, J. and Vázquez-Cendón, M.-E.: Depth averaged modelling of turbulent shallow water flow with wet-dry fronts, Arch. Comput. Methods Eng., 14(3), 303–341, doi:10.1007/s11831-007-9009-3, 2007. Dent, J. D. and Lang, T. E.: Modeling of Snow flow, J. Glaciol., 26(94), 131–140, doi:10.3189/S0022143000010674, 1980. Grimaldi, S., Li, Y., Walker, J. P. and Pauwels, V. R. N.: Effective Representation of River Geometry in Hydraulic Flood Forecast Models, Water Resour. Res., 54(2), 1031–1057, doi:10.1002/2017WR021765, 2018. Hutter, K., Koch, T., Plüess, C. and Savage, S. B.: The dynamics of avalanches of granular materials from initiation to runout. Part II. Experiments, Acta Mech., 109(1), 127–165, doi:10.1007/BF01176820, 1995. Jaedicke, C., Kern, M. A., Gauer, P., Baillifard, M. A. and Platzer, K.: Chute experiments on slushflow dynamics, Cold Reg. Sci. Technol., 51(2–3), 156–167, doi:10.1016/j.coldregions.2007.03.011, 2008. López, D., Marivela, R. and Garrote, L.: Smoothed particle hydrodynamics model applied to hydraulic structures: a hydraulic jump test case, J. Hydraul. Res., 48(sup1), 142–158, doi:10.1080/00221686.2010.9641255, 2010. López Gómez, D., Cuellar Moro, V. and Díaz Martínez, R.: Corrección termodinámica de la difusión numérica del método W-SPH, Ing. del agua, 19(1), 1, doi:10.4995/ia.2015.3140, 2015. Naaim, M., Naaim-Bouvet, F., Faug, T. and Bouchet, A.: Dense snow avalanche modeling: Flow, erosion, deposition and obstacle effects, Cold Reg. Sci. Technol., 39(2–3), 193–204, doi:10.1016/j.coldregions.2004.07.001, 2004. Naaim, M., Durand, Y., Eckert, N. and Chambon, G.: Dense avalanche friction coefficients: Influence of physical properties of snow, J. Glaciol., 59(216), 771–782, doi:10.3189/2013JoG12J205, 2013. Platzer, K., Bartelt, P. and Jaedicke, C.: Basal shear and normal stresses of dry and wet snow avalanches after a slope deviation, Cold Reg. Sci. Technol., 49(1), 11–25, doi:10.1016/j.coldregions.2007.04.003, 2007a. Platzer, K., Bartelt, P. and Kern, M.: Measurements of dense snow avalanche basal shear to normal stress ratios (S/N), Geophys. Res. Lett., 34(7), 1–5, doi:10.1029/2006GL028670, 2007b. Pudasaini, S. P. and Krautblatter, M.: Journal of Geophysical Research : Earth Surface A two-phase mechanical model for rock-ice avalanches, , 2272–2290, doi:10.1002/2014JF003183.Received, 2014. Salm, B.: A short and personal history of snow avalanche dynamics, Cold Reg. Sci. Technol., 39(2–3), 83–92, doi:10.1016/j.coldregions.2004.06.004, 2004. Sampl, P. and Granig, M.: Avalanche simulation with SAMOS-AT, ISSW 09 - Int. Snow Sci. Work. Proc., (January 2009), 519–523, 2009. Sanz-Ramos, M., Amengual, A., Bladé, E., Romero, R. and Roux, H.: Flood forecasting using a coupled hydrological and hydraulic model (based on FVM) and high resolution meteorological model, edited by A. Paquier and N. Rivière, E3S Web Conf., 40(06028), doi:10.1051/e3sconf/20184006028, 2018. Schraml, K., Thomschitz, B., Mcardell, B. W., Graf, C. and Kaitna, R.: Modeling debris-flow runout patterns on two alpine fans with different dynamic simulation models, Nat. Hazards Earth Syst. Sci., 15(7), 1483–1492, doi:10.5194/nhess-15-1483-2015, 2015.

[Figure]

Figure 1. Photography taken few days after the event from the right side of the propagation area, in the upper hillslope of the road (a). The same image, where some aspects of the area and the avalanche have been highlighted.

**Fig. 1.**